# A non-canonical visual cortical-entorhinal pathway contributes to spatial navigation

Qiming Shao [1], Ligu Chen[1], Xiaowan Li[1], Miao Li[1], Hui Cui[1], Xiaoyue Li[1], Xinran Zhao[1], Yuying Shi[1], Qiang Sun[1], Kaiyue Yan[1] & Guangfu Wang [1] ✉

Visual information is important for accurate spatial coding and memory-guided navigation. As a crucial area for spatial cognition, the medial entorhinal cortex (MEC) harbors diverse spatially tuned cells and functions as the major gateway relaying sensory inputs to the hippocampus containing place cells. However, how visual information enters the MEC has not been fully understood. Here, we identify a pathway originating in the secondary visual cortex (V2) and directly targeting MEC layer 5a (L5a). L5a neurons served as a network hub for visual processing in the MEC by routing visual inputs from multiple V2 areas to other local neurons and hippocampal CA1. Interrupting this pathway severely impaired visual stimulus-evoked neural activity in the MEC and performance of mice in navigation tasks. These observations reveal a visual cortical-entorhinal pathway highlighting the role of MEC L5a in sensory information transmission, a function typically attributed to MEC superficial layers before.

Spatial navigation is an indispensable ability for animals and humans to survive in daily life. In the mammalian navigation network, the medial entorhinal cortex (MEC) plays a critical role. Indeed, multiple spatially tuned cell types were discovered in the MEC, including grid cells[1], head direction cells[2], border cells[3], speed cells[4] and object vector cells[5]. Meanwhile, the MEC is also the major interface between other cortical regions and the hippocampus, the most pivotal region for spatial cognition[6,7]. During navigation, sensory inputs, especially visual inputs, are believed to provide external reference and correct accumulated errors generated by the path integrator[8,9]. Studies have demonstrated that manipulations of visual cues such as landmark rotation or environmental deformation can influence the firing field of grid cells and place cells[1,10–12]. Although earlier study reported that grid cells in rats were maintained in darkness[1], recent studies found that grid cells in mice degraded quickly in darkness, as well as border cells, head direction cells and speed cells in the MEC[13,14]. Furthermore, optic flow, which provides information about linear and rotational velocity, is also expected to contribute to the firing pattern of grid cells[15,16]. These facts together imply a crucial role of the visual input to the MEC. However, it is still not fully understood how visual information enters the entorhinal network from visual areas.

The MEC has a six-layered structure and is separated into superficial and deep layers by the lamina dissecans[7,17]. In canonical view, superficial layers send cortical sensory inputs to the hippocampus primarily through the perforant pathway, whereas deep layers relay spatial and memory-related hippocampal outputs to the cortical and subcortical telencephalic structures[18,19]. In line with the model, visual information is assumed to reach superficial layers of the MEC via other associational cortices[20,21], and visual-cue-dependent activity has been revealed in the perforant pathway[22]. However, recent studies showed that deep layers of the MEC may play more complex roles. Layer 5a (L5a) cells of the MEC send axonal collaterals back to the CA1 region of the hippocampus[23], and layer 6b cells project to all sub-regions of the hippocampal formation[24]. Since deep layers of the MEC also receive extensive cortical inputs[25], they are expected to gate bidirectional cortical-hippocampal communications instead of unidirectional relaying of hippocampal outputs to the neocortex. Of note, anterograde tracing study showed that the secondary visual cortex (V2) directly projects to deep layers of the MEC[26]. However, what is the target of these visual projections and how visual inputs are further routed in the local entorhinal circuitry and to the downstream hippocampus remain unclear.

[1]HIT Center for Life Sciences, School of Life Science and Technology, Harbin Institute of Technology, Harbin 150001, China.
✉e-mail: wangguangfu@hit.edu.cn

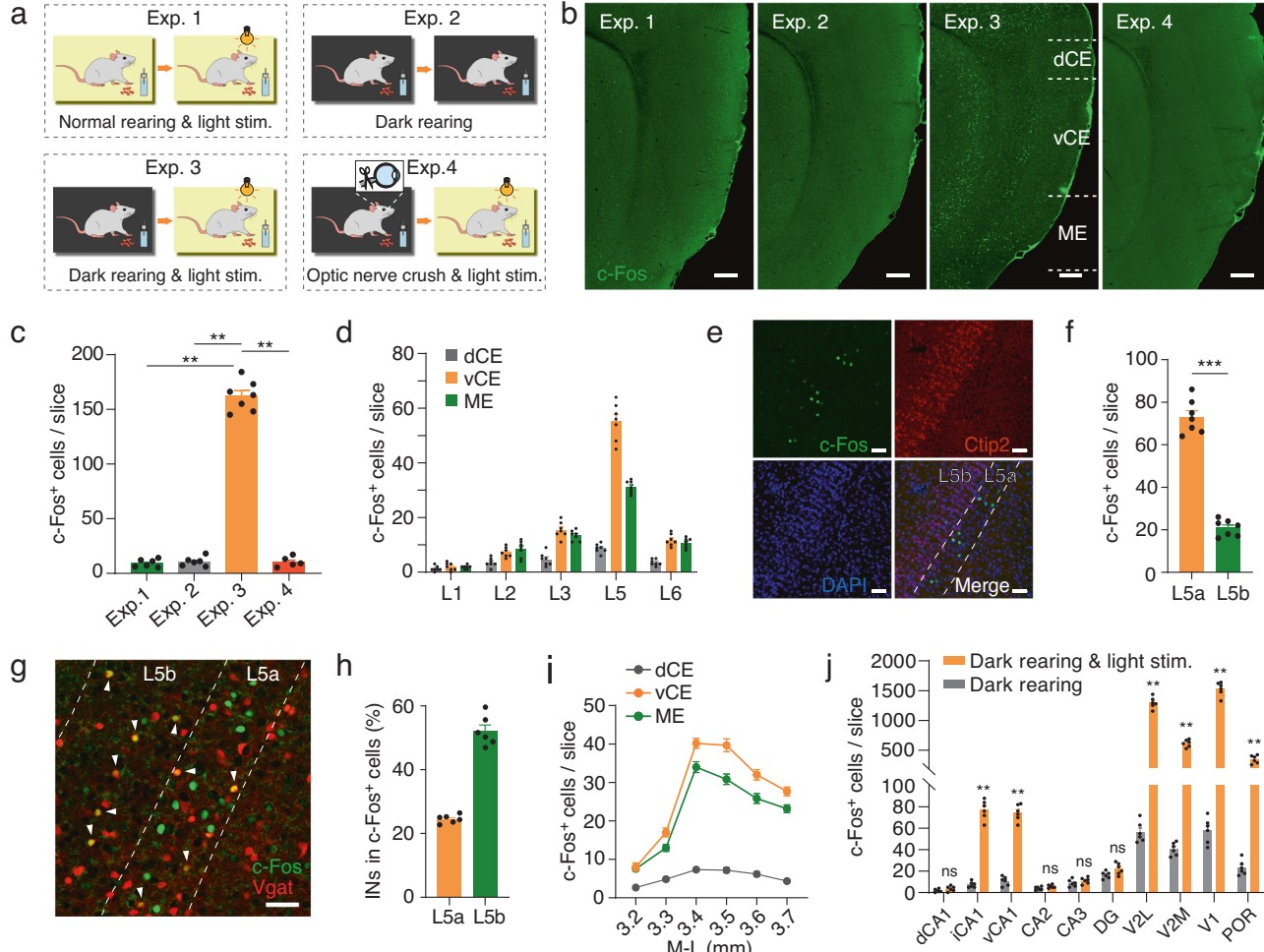

**Fig. 1 | MEC L5a responds to brief visual stimuli. a** Schematic representation of rearing and light stimulation paradigms. In experiment (Exp.) 1, 3, and 4, mice received a brief light stimulus (2 min). **b** Sagittal sections showing c-Fos expression in the MEC, which was divided into dorsal caudal (dCE), ventral caudal (vCE) and medial (ME) subdivisions along the dorsoventral (DV) axis. **c** Comparison of c-Fos⁺ cell numbers in the MEC for the mouse groups of Exp. 1-4. Cells were counted in sagittal slices within a range of 3.40–3.60 mm in the mediolateral (ML) direction. Exp. 1, $n = 6$ mice; Exp. 2, $n = 6$ mice; Exp. 3, $n = 7$ mice; Exp. 4, $n = 5$ mice. Exp. 1/2 vs. Exp. 3, **$P = 0.0012$; Exp. 4 vs. Exp. 3, **$P = 0.0025$. **d** Comparison of c-Fos⁺ cell numbers in MEC subdivisions and layers for the mouse group of Exp. 3 ($n = 7$ mice).

**e** Confocal images showing intensive c-Fos expression in MEC L5a. Note that Ctip2 is a marker of MEC L5b. **f** Comparison of c-Fos⁺ cell numbers in L5 of the MEC for the mouse group of Exp. 3 ($n = 7$ mice, ***$P = 0.0006$). **g** Confocal image showing c-Fos⁺ cells in Vgat^Cre x Ai9 mice. **h** Proportion of L5 interneurons (INs) in c-Fos⁺ cells ($n = 6$ slices from 3 mice). **i** Distribution of c-Fos⁺ cells in L5a of the MEC subdivisions along the ML direction ($n = 6$ mice). **j** Comparison of c-Fos⁺ cell numbers in the hippocampus and the visual cortical areas for the mice groups of Exp. 2 and 3 (Exp. 2, $n = 6$ mice; Exp. 3, $n = 6$ mice; **$P = 0.0022$, ns $P \geq 0.05$). Scale bars, 200 μm (**b**), 50 μm (**e** and **g**). Two-sided Mann-Whitney $U$ test (**c**, **f** and **j**). For all data, error bars represent SEM. Source data are provided as a Source Data file.

In this study, we first corroborated that neurons of the MEC responded to visual stimuli by examining *c-Fos* expression. Inspired by this finding, we used a variety of anterograde and retrograde viral tracers to investigate the projections from the visual cortex to the MEC. Intriguingly, direct visual inputs essentially originated from V2, rather than the primary visual cortex (V1), and targeted L5a neurons in the MEC. Multiple higher-order visual areas (HVAs) of the V2 were involved in this visual cortical-entorhinal pathway. By electrophysiological recordings combined with optogenetic activation, the monosynaptic nature of the projections was verified. Moreover, subsequent tracing and electrophysiological experiments showed that visual inputs from V2 were further processed along the dorsoventral (DV) axis of the MEC and routed to the hippocampal CA1 and superficial layers of the MEC. In addition, this visual cortical-entorhinal pathway was further confirmed by chemogenetic experiments combining c-Fos immunofluorescence or fiber photometry recordings, exhibiting that impaired V2 activity influenced the response of MEC L5a neurons to visual stimuli. Finally, we employed chemogenetic

inhibition of MEC L5a neurons receiving V2 projections in the Morris water maze (MWM) test or optogenetic inhibition of projecting axons from V2 in MWM and the Barnes maze (BM) tests. Both manipulations impaired the performance of animals in navigation tasks and the associated activity of MEC L5a neurons required sustained visual inputs. Our results reveal a non-canonical visual cortical-entorhinal pathway and suggest a pivotal role of this pathway in spatial navigation.

## Results

### MEC L5a responds to brief visual stimuli

Immediate early gene *c-Fos* is rapidly and transiently evoked by a wide range of external stimuli activating neurons[27,28]. To test whether the MEC responds to visual stimulation, we examined the expression of c-Fos protein in the MEC after mice received a brief light stimulus (550 lx, 2 min; Fig. 1a, Exp. 1), however, little c-Fos expression was observed (Fig. 1b, Exp. 1). Even with a much stronger light stimulus (5200 lx, 2 min), c-Fos expression remained low (Supplementary

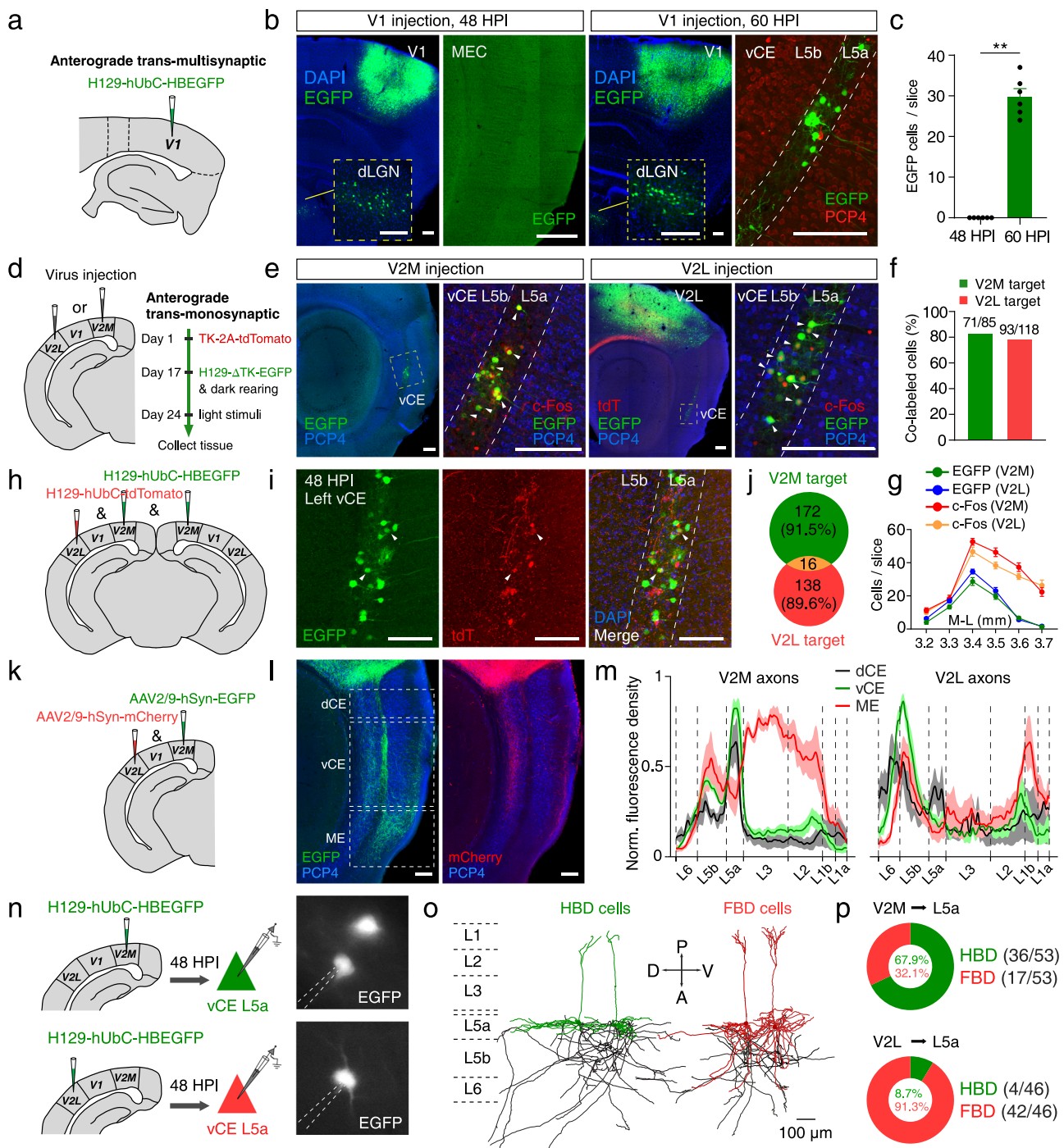

**Fig. 2 | V2 rather than V1 has monosynaptic connectivity to MEC L5a.**
**a** Schematic illustrating anterograde trans-multisynaptic HSV tracing strategy.
**b** Confocal images showing HSV1 H129 expression at 48 HPI (left) or 60 HPI (right) within the V1, dLGN (inset) and MEC. **c** Comparison of HSV1 H129 expression within MEC L5a at 48 HPI and 60 HPI (*n* = 6 mice/group). ** *P* = 0.0022, two-sided Mann-Whitney *U* test. **d** Schematic illustrating anterograde trans-monosynaptic HSV tracing and light stimulation strategy. **e** Confocal images showing H129-ΔTK-EGFP (green) and c-Fos (red) expression within the MEC when the AAV2/9-EF1α-TK-tdTomato and H129-ΔTK-EGFP were injected into V2M (left) or V2L (right).
**f** Proportion of co-labeled cells in H129-infected MEC L5a cells (V2M, *n* = 3 mice; V2L, *n* = 3 mice). **g** Distribution of c-Fos⁺ cells and HSV-expressing cells in vCE L5a along the ML direction (*n* = 3 mice). **h** Schematic illustrating injection of two kinds of HSV1 H129 within unilateral V2L and bilateral V2M, respectively. **i** Confocal

images showing the expression of the two kinds of HSV1 H129 at 48 HPI within the MEC L5a. **j** The Venn diagram shows the number of the MEC L5a cells receiving V2M projections or V2L projections (*n* = 8 slices from 4 mice). **k** Schematic illustrating injection of AAV within V2M and V2L. **l** Confocal images showing axonal fibers of V2M and V2L in the MEC. **m** Distribution of axonal fibers of V2M and V2L within the MEC (*n* = 3 mice). **n** Schematic illustrating injection of HSV1 H129 within V2M or V2L, followed by whole-cell recordings from EGFP-positive neurons of vCE L5a at 48 HPI. **o** Two morphological subtypes of MEC L5a neurons (black lines represent axons). **p** Proportions of HBD and FBD cells in MEC L5a neurons receiving V2M (*n* = 53 cells from 6 mice) or V2L (*n* = 46 cells from 5 mice) projections.
Scale bars, 100 µm (**i**), 200 µm (others). Error bars represent SEM (**c** and **g**). Shaded error bands represent SEM (**m**). Source data are provided as a Source Data file.

Fig. 1a–c). To strengthen the effect of light stimuli, we reared mice in darkness for 1–9 days before conducting the light stimuli[29,30], and then we found that c-Fos expression increased with the duration in darkness and saturated after 5 days (Supplementary Fig. 1d). Therefore, in the following study 7-day dark rearing was carried out for dark-reared mice.

Compared with those receiving no light stimuli, mice subjected to a light stimulus had a significantly higher c-Fos expression in the MEC as well as visual areas (Fig. 1a, b, Exp. 2 vs. Exp. 3; Fig. 1c and Supplementary Fig. 1h, i). Furthermore, mice with the optic nerve clipped (Supplementary Fig. 1e, f) showed tiny c-Fos response to light stimuli in the MEC (Fig. 1a, b, Exp. 4; Fig. 1c), implying that the response requires the intact visual pathway. The MEC consists of caudal (CE) and medial (ME) subdivisions[31]. Interestingly, the distribution of c-Fos expression was layer and subregion dependent, with the ventral CE and ME expressing more c-Fos proteins, especially in layer 5 (L5; Fig. 1d). We therefore counted cell numbers in the dorsal 1/4 (dCE) and the ventral 3/4 CE (vCE), respectively (Fig. 1b, Exp. 3; Supplementary Fig. 1g). With the help of Ctip2, a molecular marker of layer 5b (L5b) of the MEC[32,33], we observed that c-Fos was overwhelmingly expressed in layer 5a (L5a) rather than L5b (Fig. 1e, f). Furthermore, using CaMKII antibody or Vgat$^{Cre}$ × Ai9 transgenic mice where GABAergic interneurons (INs) are fluorescently labeled, we found that most of c-Fos positive (c-Fos$^+$) neurons in L5a were excitatory neurons and $52.0 \pm 2.0\%$ of c-Fos$^+$ neurons in L5b were INs (Fig. 1g, h and Supplementary Fig. 1k, l). In addition, along the mediolateral (ML) direction, c-Fos expression was mostly restricted to 3.40-3.60 mm lateral to the bregma (Fig. 1i and Supplementary Fig. 1j), indicating a preference for the lateral MEC.

Besides the MEC, brain-wide mapping of c-Fos expression showed that subregions of the visual cortex, the hippocampal ventral CA1 (vCA1) and intermediate CA1 (iCA1), but not dorsal CA1 (dCA1), also responded to brief visual stimuli (Fig. 1j and Supplementary Fig. 1o). In addition, sudden exposure to light did not increase c-Fos expression in the lateral (LA), basolateral (BLA) and central amygdala (CeA), brain areas associated with fear and anxiety (Supplementary Fig. 1m, n).

## V2 rather than V1 has monosynaptic connectivity to MEC L5a

Mice have a typical mammalian visual system, where the V1 receives visual inputs from the lateral geniculate nucleus (LGN) and projects to the V2[34,35]. Since MEC L5a neurons responded to a brief light stimulus, we asked whether V1 forms direct synaptic connections to MEC L5a. To answer this question, we took advantage of the high-brightness anterograde transneuronal tracer based on herpes simplex virus type 1 (HSV1) strain H129 (H129-hUbC-HBEGFP)[36]. With different hours post infection (HPI), H129-hUbC-HBEGFP can be used to dissect trans-monosynaptic or trans-multisynaptic circuits[36]. We first injected the virus into the V1 of adult wildtype (WT) mice and collected samples at 48 HPI (Fig. 2a). EGFP expression was observed at the V1 injection site and its downstream brain areas including the dorsal LGN (dLGN) and the V2L, but not in the MEC (Fig. 2b and Supplementary Fig. 2a). However, when samples were collected at 60 HPI, vCE L5a of the MEC, identified with the help of PCP4 antibody labeling L5b and L3 neurons[33], also expressed EGFP, alongside with the V1 injection site, the dLGN and the V2L (Fig. 2b, c and Supplementary Fig. 2b). Moreover, in the ML direction, EGFP-expressing neurons were mostly restricted to 3.40-3.60 mm lateral to the bregma, which is consistent with the distribution of c-Fos expression induced by brief light stimuli (Supplementary Fig. 2c, d). Given the large size of V1, we also injected H129-hUbC-HBEGFP at multiple sites of V1 and got consistent results (Supplementary Fig. 2e–g). These data were further supported by observation that the V1 hardly project to the MEC (Supplementary Fig. 2h–j). These results together imply that direct synaptic connections from V1 to MEC are rare if not

absent, but indirect connections, probably across two synapses via V2, are abundant.

To further corroborate the above results, we used the trans-monosynaptic version of HSV1 H129, i.e., H129-ΔTK-EGFP, which was generated by deleting the thymidine kinase (TK) gene and thus only labeled the postsynaptic neurons with EGFP in the present of complementarily expressed TK from a helper virus[37]. The helper virus AAV2/9-EF1α-TK-tdTomato and H129-ΔTK-EGFP were injected into V2M or V2L of WT mice on day 1 and day 17 sequentially (Fig. 2d). Mice injected with the last virus were immediately fed in darkness for 7 days, and finally given a light stimulus for 2 min to induce c-Fos expression (Fig. 2d). Neurons labeled with both tdTomato and EGFP (starter neurons) were observed at the injection site of V2M (Supplementary Fig. 3a, b) or V2L (Supplementary Fig. 3c) and postsynaptic neurons only labeled with EGFP were found in the vCE (Fig. 2e). In contrast, no MEC neuron was labeled by EGFP when AAV2/9-EF1α-TK-tdTomato and H129-ΔTK-EGFP were injected into V1 (Supplementary Fig. 2k–o). With the help of PCP4 antibody, we confirmed that EGFP-labeled neurons were almost exclusively located in vCE L5a (Fig. 2e) and a small proportion of them were INs (Supplementary Fig. 3d, e). Anterograde trans-monosynaptic tracing by H129-hUbC-HBEGFP, injected either at similar positions or more medially and anteriorly in V2L and V2M, corroborated vCE L5a as the main target (Supplementary Fig. 3g–n). Additionally, the data also showed that V2M targeted the contralateral vCE L5a while V2L targeted the contralateral dCE L5a (Supplementary Fig. 3g–j). In the ML direction, the distribution of EGFP-labeled vCE L5a neurons was again consistent with c-Fos expression induced by the light stimulus (Fig. 2g and Supplementary Fig. 3f). In fact, EGFP and c-Fos had a high co-expression rate in vCE L5a (Fig. 2f), suggesting that V2 projections are engaged in c-Fos responses to the light stimulus. Taken together, these results imply a pathway that V2 relays visual signals from V1 to vCE L5a, i.e., V1→V2→MEC L5a pathway.

Since both V2M and V2L targeted vCE L5a neurons (Fig. 2e), we asked whether V2M and V2L target the same L5a neurons. Considering that contralateral V2M also targeted a small number of vCE L5a neurons (Supplementary Fig. 3i), we injected H129-hUbC-tdTomato into V2L unilaterally but into V2M bilaterally, and found that V2M and V2L rarely targeted the same L5a neurons (Fig. 2h–j). To gain further insight into the target of V2, we examined the distribution of V2 axonal terminals in the MEC. EGFP-expressing virus was injected into V2M, and mCherry-expressing virus was injected into V2L simultaneously (Fig. 2k). Within the dCE, vCE and ME subdivisions of the MEC, abundant axon fibers from V2M or V2L were differentially distributed (Fig. 2l). In the vCE, axons from V2M were primarily found in L5a, whereas axons from V2L were primarily found in L5b (Fig. 2l, m). To further identify the postsynaptic targets of V2M and V2L projections, we injected H129-hUbC-HBEGFP into V2M or V2L and performed whole-cell recordings at 48 HPI to load biocytin into the EGFP-positive pyramidal cells (PCs) in vCE L5a for immunostaining and morphological reconstruction (Fig. 2n). The recovered morphologies exhibited that vCE L5a PCs could be largely divided into two subtypes based on the shape of their basal dendrites. One subtype had horizontal basal dendrites (HBDs) mainly restricted to L5a, whereas the other subtype had fan-shaped basal dendrites (FBDs) extending both horizontally within L5a and vertically into L5b/6 (Fig. 2o and Supplementary Fig. 4a, b). We found that V2M and V2L tended to target different subtypes of vCE L5a PCs (Fig. 2p). While V2M-targeted neurons were mostly HBD cells (67.9%), V2L-targeted neurons were dominated by FBD cells (91.3%). Interestingly, the preference of V2M for HBD cells could not be predicted directly from the distributions of projecting axons and basal dendrites. To see whether it was a genuine preference or just a reflection of the ratio between the two subtypes in all L5a PCs, we randomly recorded and reconstructed 151 L5a PCs from untreated mice and found that HBD and FBD cells accounted for 41.1% and 58.9%,

respectively (HBD: $n = 62$; FBD: $n = 89$). Thus, the ratio between the two subtypes could not explain the preference of V2M for HBD cells. Taken together, our data show that V2M and V2L have distinct spatial distributions of axons in the MEC, which have preference for targeting different neuronal populations with distinct morphologies.

### Multiple higher-order visual areas contribute to the V1→V2→MEC L5a pathway

Although rodents have much simpler HVAs than primates, subareas of V2 have been identified, including lateromedial (LM), laterointermediate (LI), anterolateral (AL), posterior (P), anterior (A), rostrolateral (RL), anteromedial (AM) and posteromedial (PM) areas[38–40]. Therefore, we next investigated how these subareas are involved in V1→V2→MEC L5a pathway. MEC L5a is a major output layer projecting to diverse subcortical and cortical structures including V2M[23,32,41]. To specifically target MEC L5a neurons as well as to examine reciprocal projections between V2 and MEC L5a, we simultaneously injected retrograde tracers, AAV2retro-CAG-mCherry and AAV2retro-CAG-EGFP, into V2L and V2M, respectively (Fig. 3a). As reported in previous studies, MEC L5a neurons targeting V2M were observed[32,41]. Meanwhile, MEC L5a neurons targeting V2L were also present (Fig. 3b). Throughout the DV axis of the MEC, L5a neurons targeting V2L or V2M were distributed (Fig. 3b, c), and some of them targeted both regions as indicated by double labeling (Fig. 3b).

Based on the above results, we combined AAV2retro and rabies virus-based retrograde trans-monosynaptic tracer[42] to trace the presynaptic neurons innervating MEC L5a. Briefly, on day 1 V2L and V2M were injected with AAV2retro-CAG-Cre, and MEC L5a neurons were injected with AAV driving the Cre-dependent expression of the avian TVA receptor and the rabies glycoprotein (helper virus). Next, on day 18 RV-CVS-ENVA-N2C(ΔG)-tdTomato was injected into MEC L5a to infect Cre-labeled helper-positive L5a neurons. Finally, on day 24 brain tissues were collected (Supplementary Fig. 5a). In sagittal sections, double-labeled neurons (starter neurons) were found in MEC L5a as expected (Supplementary Fig. 5b). Consistent with anterograde tracing data, presynaptic neurons in V2L and V2M, rather than V1, were observed (Supplementary Fig. 5b–d). After validating the RV tracing, we also injected anterograde tracer H129-hUbC-HBEGFP into V1 two days before tissue collection (Fig. 3d). In tangential sections through flat-mounted cortex, double-labeled neurons present in MEC L5a (Fig. 3e) were putative starter neurons since injection of HSV1 H129 in V1 did not label MEC cells at 48 HPI (Figs. 2b, c; Supplementary Fig. 2e–g). Meanwhile, presynaptic neurons of MEC L5a were observed to locate topographically in areas LM, LI, P, AL, AM, PM and the postrhinal cortex (POR), but not in RL or A (Fig. 3f, g). Of note, RV-tdTomato and HSV-EGFP co-expressing neurons were found in LM, AL and AM (Fig. 3f), indicating that the same V2 neurons receiving V1 projections can innervate MEC L5a directly. Moreover, a lot of other brain areas innervating MEC L5a were identified, including MEC L3, MEC L5b, subiculum (Sub), dCA1, iCA1, vCA1, the retrosplenial cortex (RSC), the claustrum (Cl), the anterodorsal thalamic nucleus (ADT) and the anteromedial thalamic nucleus (AMT; Fig. 3h–m). However, none of these areas occupy a position prior to V2 in the hierarchy of visual processing. Together, these data provide direct evidence for the V1→V2→MEC L5a pathway and demonstrate the involvement of multiple V2 subareas in it.

### V2 inputs evoke excitatory and inhibitory synaptic responses in MEC L5a neurons

To corroborate that V2→MEC L5a pathway is mediated by functional synapses, we delivered AAV2/9-CaMKII-ChR2-mCherry into V2M or V2L of WT mice (Fig. 4a, e). We then activated V2 projections in brain slices by light pulses and recorded postsynaptic currents (PSCs) from MEC neurons. Optical activation of afferent fibers from V2M or V2L

evoked excitatory PSCs (EPSCs) in a large proportion of vCE L5a PCs, but in none of neurons of L5b and L2/3 (Fig. 4b, c and Fig. 4f, g). In addition, light-evoked EPSCs in vCE L5a PCs were abolished after bath application of tetrodotoxin (TTX) but were then reintroduced following application of 4-Aminopyridine (4-AP; Fig. 4d and h left; Fig. 4i), indicating the monosynaptic nature of connections from V2 to vCE L5a. These restored responses were blocked by AMPA receptor antagonist CNQX (Figs. 4d, 4h and 4i), indicating that the connections are glutamatergic. When vCE L5a PCs were clamped at 0 mV, inhibitory PSCs (IPSCs) were recorded. They were also abolished by TTX but were not recovered by 4-AP (Fig. 4d, h right; Fig. 4j). Furthermore, the onset latencies of IPSCs were twice longer than those of EPSCs (Fig. 4k), indicating that the IPSCs were due to polysynaptic inhibition mediated by local inhibitory neurons. The amplitudes of EPSCs and IPSCs evoked by V2M were significantly larger than those evoked by V2L (Fig. 4l), possibly because the projecting fibers from V2M were densely packed in vCE L5a whereas those from V2L were mainly distributed in vCE L5b (Fig. 2l–m).

### Downstream targets of the V2→MEC L5a pathway in the MEC and hippocampus

Hitherto, we had confirmed the V2→MEC L5a pathway by anterograde and retrograde tracing as well as optogenetics. We wondered how the inputs through this pathway were further delivered in local circuits. To this end, we injected H129-hUbC-HBEGFP into V2L of adult WT mice and collected samples at 60 HPI. A lot of EGFP-positive neurons were observed in vCE L5b and L6, but surprisingly PCP4-positive neurons were rare, implying that most of them were INs (Supplementary Fig. 6a, b). Thus, we injected H129-hUbC-HBEGFP into V2L of adult Vgat[Cre] x Ai9, SOM[Cre] x Ai9 or PV[Cre] x Ai9 mice and collected samples at 48 HPI or 60 HPI (Fig. 5a). As expected, EGFP-positive neurons were observed only in vCE L5a at 48 HPI, and then showed up in vCE L5b/6 at 60 HPI (Fig. 5b, c). In these transgenic mouse strains, we found that almost all EGFP-positive cells in vCE L5b/6 were INs (Fig. 5b middle and 5d). $74.6 \pm 1.6\%$ of them were somatostatin-positive (SOM-positive) (Fig. 5b right and d), and $2.8 \pm 0.1\%$ of them were parvalbumin-positive (PV-positive) (Fig. 5d and Supplementary Fig. 6c). At 60 HPI, a small number of EGFP-positive neurons were also found in superficial layers of the MEC and hippocampal CA1. Most EGFP-positive neurons in MEC L3 were also INs, whereas those in MEC L2 and CA1 were rarely INs (Fig. 5c and Supplementary Fig. 6d, e). Furthermore, when H129-hUbC-HBEGFP was injected in V2M, largely similar results were obtained, namely $94.8 \pm 0.5\%$ of EGFP-positive cells in vCE L5b/6 were INs and $52.7 \pm 1.6\%$ were SOM-positive (Supplementary Fig. 6f–h). In addition, within vCE L5a the proportion of INs in EGFP-positive cells also increased significantly at 60 HPI (48 HPI: $1.0 \pm 0.5\%$; 60 HPI: $19.6 \pm 0.8\%$; $P < 0.01$; Fig. 5e), revealing connections from excitatory neurons to INs within L5a. In the ML direction, the distributions of EGFP-labeled L5a neurons at 48 HPI and EGFP-labeled L5b/6 neurons at 60 HPI were highly correlated, and were consistent with the distributions of c-Fos expression induced by light stimulus and EGFP expression driven by H129-ΔTK-EGFP (Fig. 5f; Figs. 1i and 2g), providing further support for the idea that these EGFP-labeled L5b/6 INs were innervated by L5a neurons receiving V2 inputs.

Since SOM INs dominated the L5b/6 targets of vCE L5a neurons receiving V2 inputs, we next employed optogenetics and whole-cell recordings to further scrutinize these connections. We first injected AAV with SOM promoters, AAV2/9-SOM-EGFP, into MEC of adult SOM[Cre] x Ai9 mice (Supplementary Fig. 7a). Our data showed that $90.9 \pm 0.7\%$ of EGFP-positive neurons were co-labeled with tdTomato (Supplementary Fig. 7b), indicating a specific labeling for SOM INs. Therefore, in the following experiments, AAV2/9-SOM-EGFP was used to identify SOM INs in WT mice. Meanwhile, we took advantage of the anterograde trans-synaptic tracer AAV1-hSyn-NLS-Cre to conduct conditional expression in postsynaptic neurons[43]. By injecting AAV1-

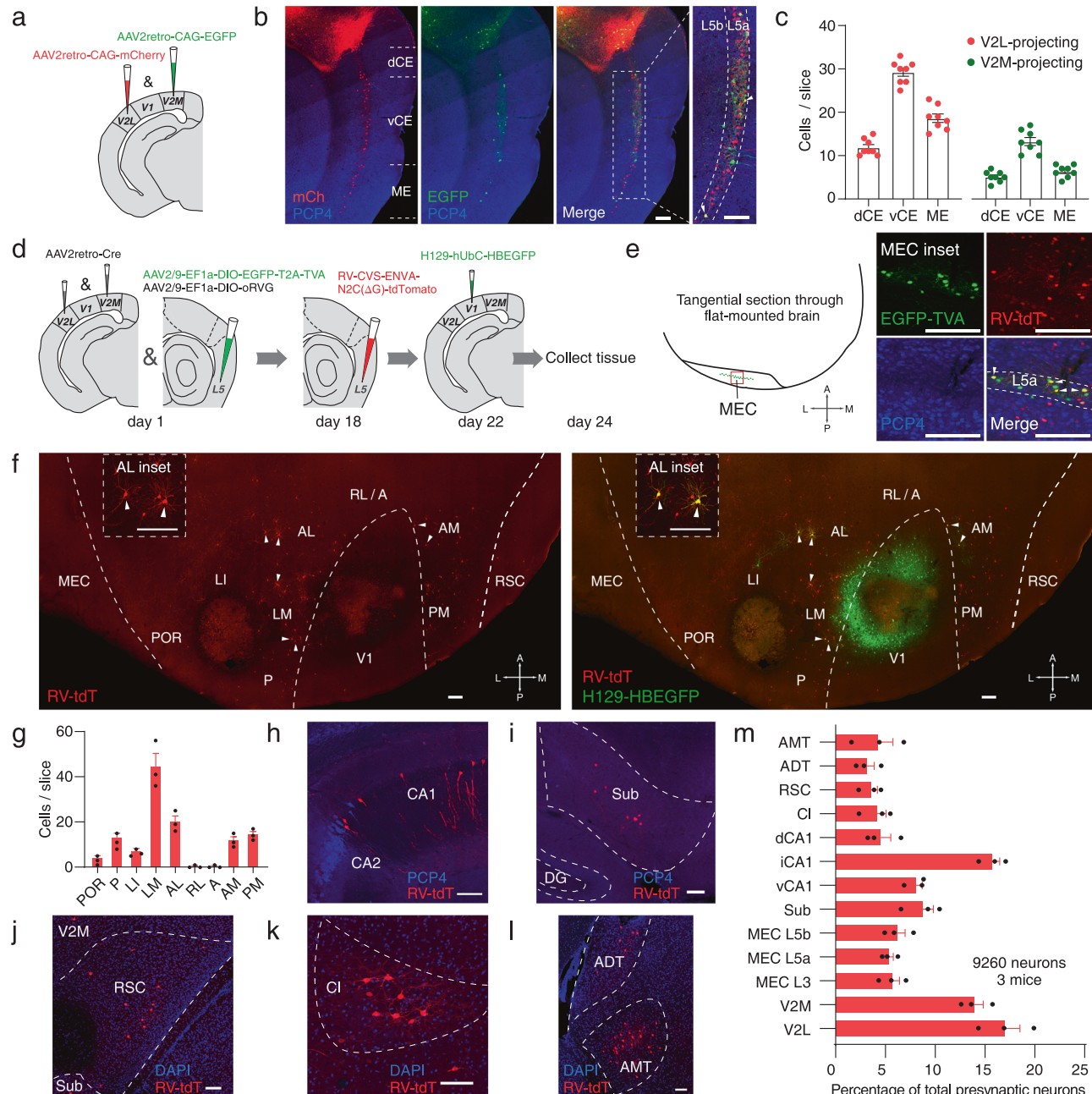

**Fig. 3 | Multiple higher-order visual areas contribute to the V1→V2→MEC L5a pathway. a** Schematic illustrating retrograde AAV tracing strategy. Two kinds of AAV2retro were injected within V2M and V2L, respectively. **b** Sagittal section showing the expression of retrograde AAV in the MEC. Arrowheads show the L5a cells targeting both V2L and V2M. **c** Comparison of retrograde AAV expression within dCE, vCE and ME L5a ($n = 8$ slices from 4 mice). **d** Schematic illustrating strategy of retrograde trans-monosynaptic RV tracing from MEC and anterograde trans-monosynaptic HSV tracing from V1. **e** Left: schematic representation of a tangential section with helper virus and RV expressed in the MEC. Right: confocal images showing viral expression within the MEC. Starter cells (yellow, arrowheads) co-expressed AAV-DIO-EGFP-TVA (green) and RV-CVS-ENVA-N2C(ΔG)-tdTomato (red). **f** Confocal images showing RV-labeled neurons (red) and HSV-labeled

neurons (green) in visual cortical areas. Inset: cells co-expressing RV-CVS-ENVA-N2C(ΔG)-tdTomato and HSV-H129-EGFP (yellow, arrowheads) in AL. A anterior, AL anterolateral, AM anteromedial, LI laterointermediate, LM lateromedial, P posterior, PM posteromedial, POR postrhinal cortex, RL rostrolateral. **g** Quantification of RV-labeled neurons in the HVAs ($n = 3$ mice). **h–l** Representative images showing retrogradely labeled presynaptic neurons of MEC L5a in brain areas other than the visual cortex. Sub subiculum, DG dentate gyrus, RSC retrosplenial cortex, Cl claustrum, ADT anterodorsal thalamic nucleus, AMT anteromedial thalamic nucleus. **m** Distribution of brain-wide input neurons to MEC L5a ($n = 9260$ neurons from 3 mice). Scale bars, 200 μm. Error bars represent SEM (**c, g** and **m**). Source data are provided as a Source Data file.

hSyn-NLS-Cre into V2L and V2M, and AAV-DIO-CaMKII-ChR2-mCherry into MEC L5, we drove the expression of ChR2 in vCE L5a neurons receiving V2 inputs (Fig. 5g and Supplementary Fig. 7c). These ChR2-mCherry expressing neurons could be activated to generate action potentials (APs) by trains of blue light pulses (5 pulses at 10 Hz; Supplementary Fig. 7d). We then recorded putative SOM INs expressing

EGFP in L5b as well as in L5a (Fig. 5h, i). The firing pattern confirmed that the recorded cells were not principal neurons (Fig. 5h, i). In some EGFP-expressing neurons, responses to light activation of L5a ChR2-expressing neurons were detected (Fig. 5h, i; Supplementary Fig. 7e–g). The light-evoked EPSCs were abolished by TTX and rescued by 4-AP, but severely attenuated by CNQX (Fig. 5h, i), indicating the

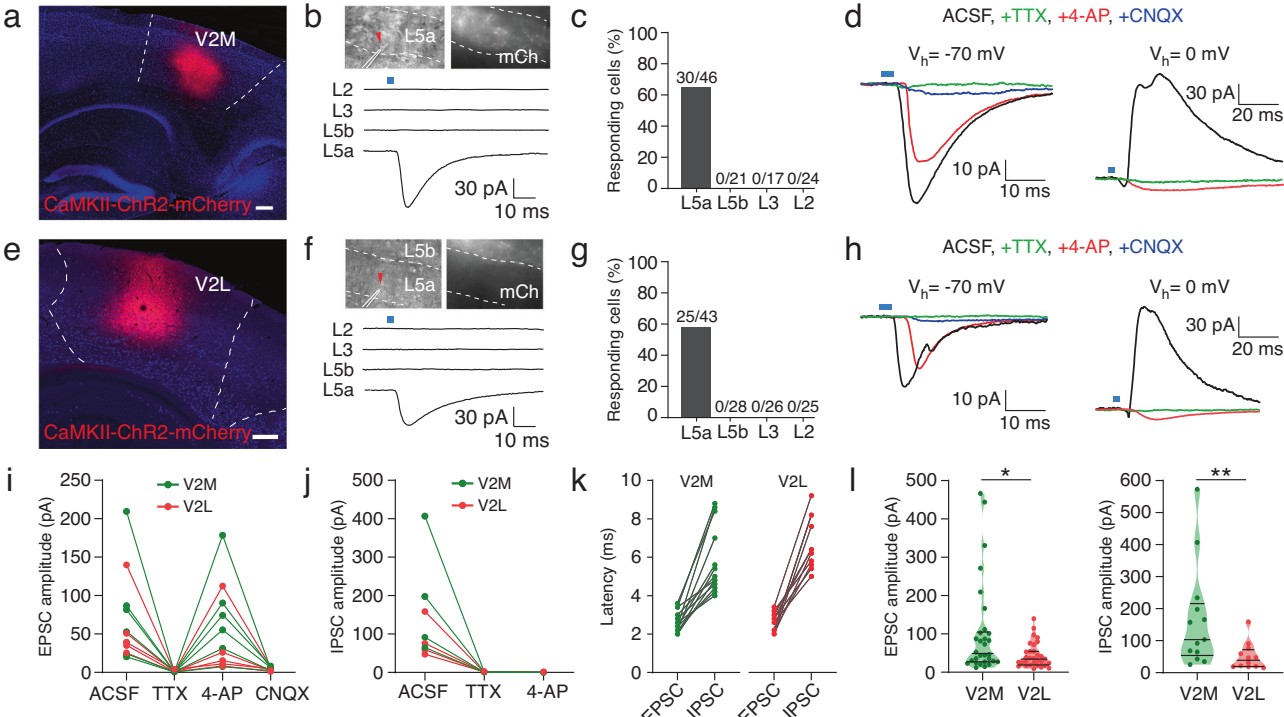

**Fig. 4 | V2 inputs evoke excitatory and inhibitory synaptic responses in MEC L5a neurons. a, e** Confocal images showing the expression of AAV2/9-CaMKII-ChR2-mCherry in V2M (**a**) or V2L (**e**). Scale bars, 200 μm. **b, f** Top: widefield images under transmitted light (left) and fluorescence (right) microscopy showing the position of a recorded cell (red arrowheads) and ChR2-mCherry expressing fibers from V2M (**b**) or V2L (**f**). Bottom: representative traces displaying responses of neurons in different layers of the MEC to optical activation of ChR2-mCherry expressing fibers. The blue bars represent light stimulation pulse (3 ms). **c, g** Proportion of cells responding to optical activation of ChR2-mCherry expressing fibers from V2M (**c**) or V2L (**g**) in each layer of the MEC. **d, h** Representative traces exhibiting effects of pharmacological blockers on EPSCs (left) and IPSCs (right) recorded from L5a neurons following activation of ChR2-mCherry expressing fibers from V2M (**d**) or V2L (**h**). **i** Summary plot exhibiting effects of pharmacological blockers on EPSCs recorded from V2M- and V2L-targeted MEC L5a neurons (V2M, $n = 6$ cells from 5 mice; V2L, $n = 5$ cells from 5 mice). **j** Summary plot exhibiting effects of pharmacological blockers on IPSCs recorded from V2M- and V2L-targeted MEC L5a neurons (V2M, $n = 4$ cells from 4 mice; V2L, $n = 4$ cells from 4 mice). **k** Summary plot exhibiting EPSC latencies vs. IPSC latencies recorded in MEC L5a neurons following activation of projections from V2M (left, $n = 13$ cells from 7 mice) or V2L (right, $n = 10$ cells from 6 mice). **l** Left: EPSC amplitude recorded from MEC L5a neurons following activation of V2M ($n = 30$ cells from 7 mice) or V2L ($n = 31$ cells from 7 mice). Right: IPSC amplitude recorded from MEC L5a neurons following activation of V2M ($n = 13$ cells from 7 mice) or V2L ($n = 12$ cells from 6 mice), *$P = 0.0315$, **$P = 0.0055$, two-sided Mann-Whitney U test. Bars represent the medians and the quartiles. Source data are provided as a Source Data file.

monosynaptic and glutamatergic nature of the connections. Moreover, in vCE L5a PCs not expressing EGFP and mCherry, light-evoked responses were also observed (Fig. 5j and Supplementary Fig. 7h). Further pharmacological operations indicated that L5a PCs receiving V2 inputs could evoke monosynaptic EPSCs and polysynaptic IPSCs in other L5a PCs (Fig. 5j). Besides, hippocampal CA1 and superficial MEC are two critical areas for spatial navigation[1,44]. Our above tracing data had showed that neurons thereof were also targeted by vCE L5a neurons receiving V2 inputs (Fig. 5c and Supplementary Fig. 6d, e). Therefore, we also performed optogenetic stimulation and whole-cell recordings in these areas. Again, we observed light-evoked responses in CA1 PCs and MEC L2 neurons (Fig. 5k, l and Supplementary Fig. 7h). Pharmacological operations indicated that vCE L5a neurons receiving V2 inputs could excite CA1 and MEC L2 neurons directly and inhibit the latter by recruiting local inhibitory circuits (Fig. 5k, l and Supplementary Fig. 7i). Taken together, these results reveal that vCE L5a neurons function as the hub to distribute V2 inputs further to their surrounding INs, especially SOM INs, as well as hippocampal CA1 and superficial MEC.

## MEC L5a routes visual information to the hippocampus and superficial MEC through the DV pathway

After confirming the projections from MEC L5a neurons receiving V2 inputs to the hippocampus and superficial MEC, we wondered how these L5a neurons were organized in the MEC. To this end, we injected retrograde virus AAV2retro-CAG-mCherry into MEC L2/3 or hippocampal iCA1/vCA1 on day 1 and anterograde trans-multisynaptic virus H129-hUbC-HBEGFP into V1 on day 18, and then collected samples on day 20.5 or 21, i.e., 60 HPI or 72 HPI of HSV (Fig. 6a). Consistent with the previous results (Fig. 2a–c), at 60 HPI most of the EGFP-labeled L5a neurons distributed in the vCE (Fig. 6b₁, c₁ and d₁). However, the MEC L2/3-projecting and iCA1/vCA1-projecting neurons of L5a were mainly found in the ME (Fig. 6b₁, c₁ and d₁). In particular, projecting fibers from ME L5a neurons were observed in superficial layers of the dorsal MEC (Supplementary Fig. 8a, b). The mismatch between the distributions of visual input receivers and senders in MEC L5a (Fig. 6e) may partially explain the relatively small number of superficial MEC and CA1 neurons targeted by MEC L5a neurons receiving V2 projections (Fig. 5c). Throughout the vCE and ME, a smaller number of co-labeled L5a neurons receiving visual inputs and projecting directly to superficial MEC and CA1 were observed (Fig. 6b₁, c₁ and d₁), and their distributions were essentially defined by the distributions of receivers and senders (Fig. 6b₃, c₃ and d₃; green curves). On the other hand, at 72 HPI EGFP spread profusely to ME L5a (Fig. 6b₂, c₂ and d₂). A large proportion of EGFP-expressing L5a neurons were also mCherry-expressing MEC L2/3- or iCA1/vCA1-projecting neurons (Fig. 6b₃, c₃ and d₃; red curves). Together, these data suggest a V2→vCE L5a→ME L5a→superficial MEC/hippocampus pathway, through which visual information is finally routed to the key areas for spatial navigation.

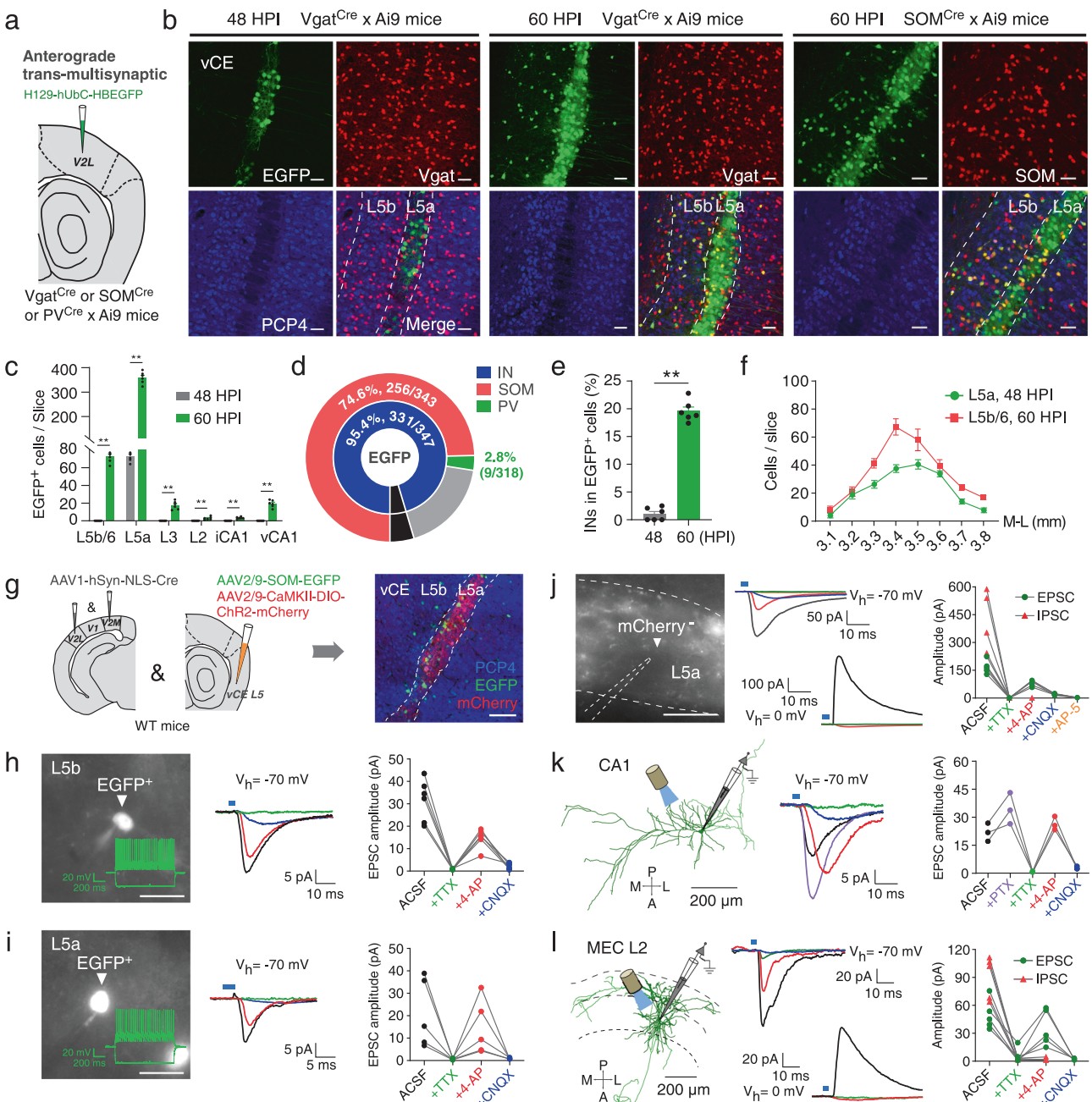

**Fig. 5 | Downstream targets of V2→MEC L5a pathway in MEC and hippocampus.**
**a** Schematic illustrating experimental strategy. **b** Confocal images showing vCE L5b neurons expressing H129-EGFP (green) at 48/60 HPI in the indicated transgenic mice. **c** Comparison of HSV1 H129 expression within hippocampal CA1 and MEC layers at 48/60 HPI ($n = 6$ mice/group; **$P = 0.0022$). **d** Proportions of INs in EGFP-positive cells within MEC L5b/6 ($n = 4$ mice/group). **e** Proportion of INs in MEC L5a EGFP-positive cells at 48/60 HPI ($n = 6$ mice/group; **$P = 0.0022$). **f** ML distributions of HSV1 H129 expression in L5a at 48 HPI and in L5b/6 at 60 HPI ($n = 4$ mice/group). **g** Left: schematic illustrating experimental strategy. Right: confocal image showing ChR2-positive neurons (red) and SOM INs (green). **h–i** Left: widefield images showing the whole-cell recording of EGFP-positive neurons in vCE L5b (**h**) or L5a (**i**). Inset: voltage responses of the recorded neurons to −200/+140 pA current pulse. Right: representative traces and summary plot exhibiting effects of

pharmacological blockers on EPSCs recorded from EGFP-positive neurons in vCE L5b (**h**; $n = 6$ cells from 5 mice) or L5a (**i**; $n = 5$ cells from 4 mice). **j** Left: widefield image showing the whole-cell recording of mCherry-negative neurons in vCE L5a. Right: representative traces and summary plot exhibiting effects of pharmacological blockers on EPSCs ($n = 5$ cells from 5 mice) or IPSCs ($n = 4$ cells from 4 mice) recorded from vCE L5a mCherry-negative neurons. **k–l** Left: schematic illustrating whole-cell recordings from hippocampal CA1 PCs (**k**) or MEC L2 neurons (**l**; axons in light color). Right: representative traces and summary plot exhibiting effects of pharmacological blockers on PSCs recorded from CA1 PCs (**k**; $n = 3$ cells from 3 mice) or MEC L2 neurons (**l**; EPSCs, $n = 5$ cells from 5 mice; IPSCs, $n = 5$ cells from 4 mice). Scale bars, 100 μm (**g**), 50 μm (others). Two-sided Mann-Whitney $U$ test (**c** and **e**). Error bars represent SEM (**c**, **e** and **f**). Source data are provided as a Source Data file.

At 72 HPI of HSV in V1, the EGFP-expressing L5b/6 neurons, largely restricted to the vCE, were observed (Fig. 6b₂, c₂, d₂, f), consistent with the results of 60 HPI in V2 (Fig. 5b and Supplementary Fig. 6a–c). As already demonstrated, these vCE L5b/6 neurons were almost exclusively INs and were dominated by SOM INs (Fig. 5d and

Supplementary Fig. 6h). We therefore wondered whether vCE L5b/6 neurons, especially SOM INs, targeted by the vCE L5a neurons receiving V2 inputs also innervate ME L5a. To this end, we combined AAV2retro and rabies virus-based retrograde trans-monosynaptic tracer to trace the presynaptic neurons innervating ME L5a

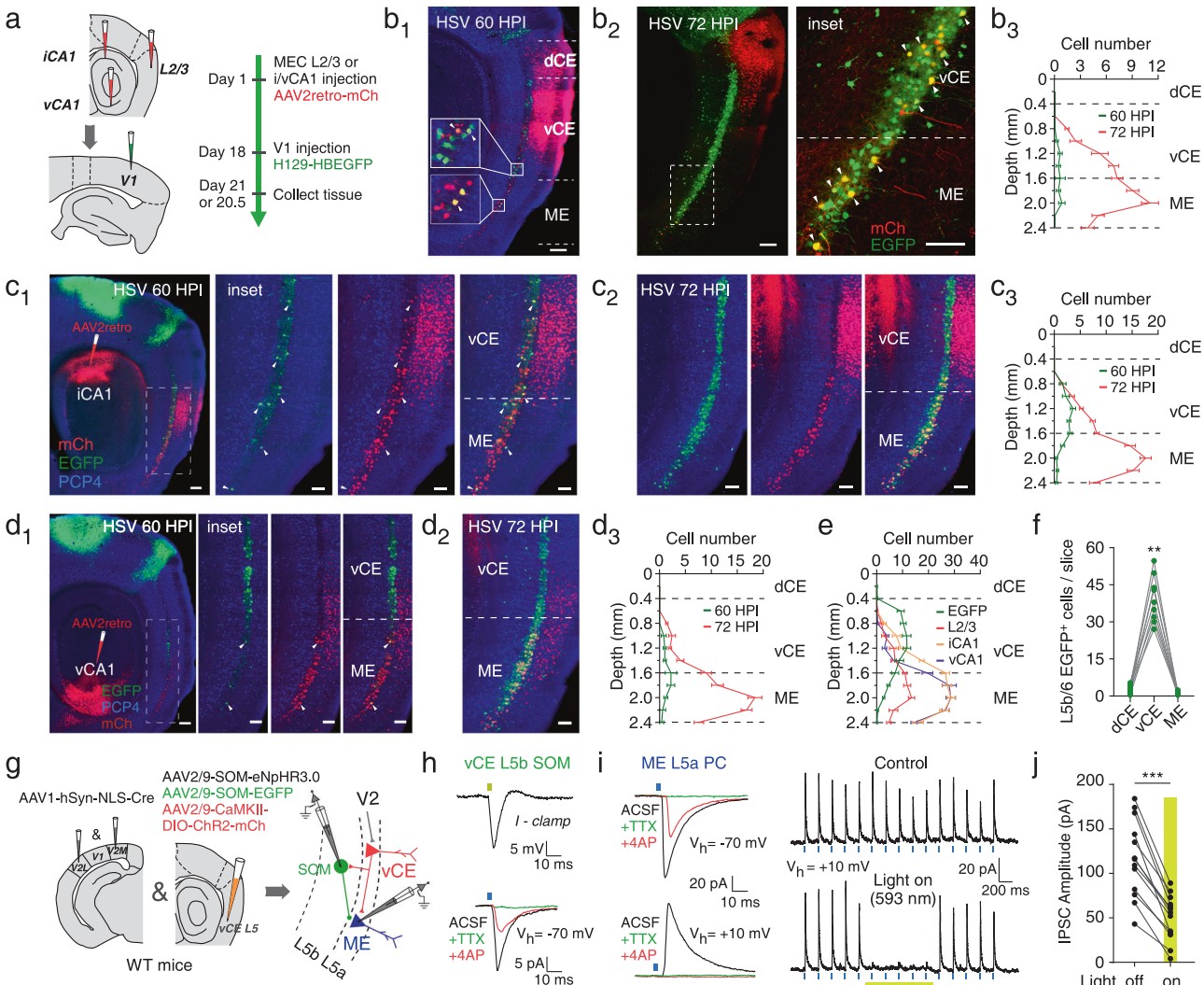

**Fig. 6 | MEC L5a routes visual information to the hippocampus and superficial MEC through the DV pathway. a** Schematic illustrating experimental strategy. **b₁–₃** Confocal images showing the expression of retrograde AAV (red) injected at MEC L2/3 and trans-multisynaptic anterograde HSV (green) at 60 (**b₁**) or 72 HPI (**b₂**). Distribution of L5a neurons co-expressing AAV and HSV along the DV axis at 60 and 72 HPI (**b₃**; *n* = 6 slices from 3 mice/group). **c₁–₃** Confocal images showing the expression of retrograde AAV (red) injected at iCA1 and trans-multisynaptic ante-rograde HSV (green) at 60 (**c₁**) or 72 HPI (**c₂**). Distribution of L5a neurons co-expressing AAV and HSV along the DV axis at 60 and 72 HPI (**c₃**; *n* = 6 slices from 3 mice/group). **d₁–₃** Confocal images showing the expression of retrograde AAV (red) injected at vCA1 and trans-multisynaptic anterograde HSV (green) at 60 (**d₁**) or 72 HPI (**d₂**). Distribution of L5a neurons co-expressing AAV and HSV along the DV axis at 60 and 72 HPI (**d₃**; *n* = 6 slices from 3 mice/group). **e** Distribution of MEC L2/3-

projecting, iCA1-projecting, vCA1-projecting and EGFP-positive L5a neurons along the DV axis at 60 HPI (*n* = 6 slices from 3 mice/group). **f** Comparison of L5b/6 EGFP-positive neuron numbers within MEC subdivisions (*n* = 9 mice; **P = 0.0039). **g** Schematic illustrating experimental strategy. **h** Representative traces exhibiting responses of vCE L5b SOM INs to optical stimulation by yellow and blue pulses. **i** Left: representative traces exhibiting effects of pharmacological blockers on EPSCs (top) and IPSCs (bottom) recorded from ME L5a PCs following activation of ChR2-mCherry expressing fibers from vCE L5a PCs. Right: representative traces exhibiting effects of inhibiting SOM INs with yellow light (1 s) on blue light-evoked IPSCs (5 Hz). **j** Comparison of IPSC amplitudes of ME L5a PCs with yellow light off and on (*n* = 12 cells from 3 mice; ***P = 0.0002). Scale bars, 100 μm (**b₁, b₂** left, **c₁** left and **d₁** left), 50 μm (others). Two-sided Wilcoxon signed-rank test (**f** and **j**). Error bars represent SEM (**b–e**). Source data are provided as a Source Data file.

(Supplementary Fig. 8c). As expected, presynaptic neurons of ME L5a were found in vCE L5a and L5b, and a significant proportion (18.9 ± 2.0 %) of presynaptic neurons in vCE L5b were SOM INs (Supplementary Fig. 8d, e). Next, we expressed the inhibitory opsin eNpHR3.0 in vCE L5 SOM INs and the excitatory opsin ChR2 in the vCE L5a neurons receiving V2 inputs (Fig. 6g and Supplementary Fig. 8f). Blue light pulses evoked monosynaptic EPSCs in vCE L5 SOM INs and ME L5a PCs, as well as polysynaptic IPSCs in the latter (Fig. 6h, i). Further-more, these polysynaptic IPSCs were suppressed by yellow light-induced inhibition of SOM INs (Figs. 6i, j), indicating a substantial contribution of SOM INs in inhibiting ME L5a PCs. Additionally, EPSCs in ME L5a PCs were enhanced during the inhibition of SOM INs (Supplementary Fig. 8g, h). These results together indicate that vCE L5

SOM INs are engaged in the dorsoventral pathway by mediating feedforward inhibition from the V2-targeted vCE L5a neurons to ME L5a.

## The V2→MEC L5a pathway is critical for spatial navigation

So far, we had identified the V2→MEC L5a pathway and assumed it to underlie the c-Fos expression enhancement in the MEC induced by visual stimuli. To validate the assumption, we tested whether inacti-vation of V2 affects the c-Fos response of MEC L5a neurons to visual stimuli. We bilaterally delivered a virus encoding inhibitory DREADD hM4Di (AAV2/9-hM4Di-EGFP) into both V2L and V2M (Fig. 7a, b and Supplementary Fig. 9a). In the dark rearing phase, hM4Di ligand clozapine-N-oxide (CNO, 10 mg/L) was delivered to mice via drinking

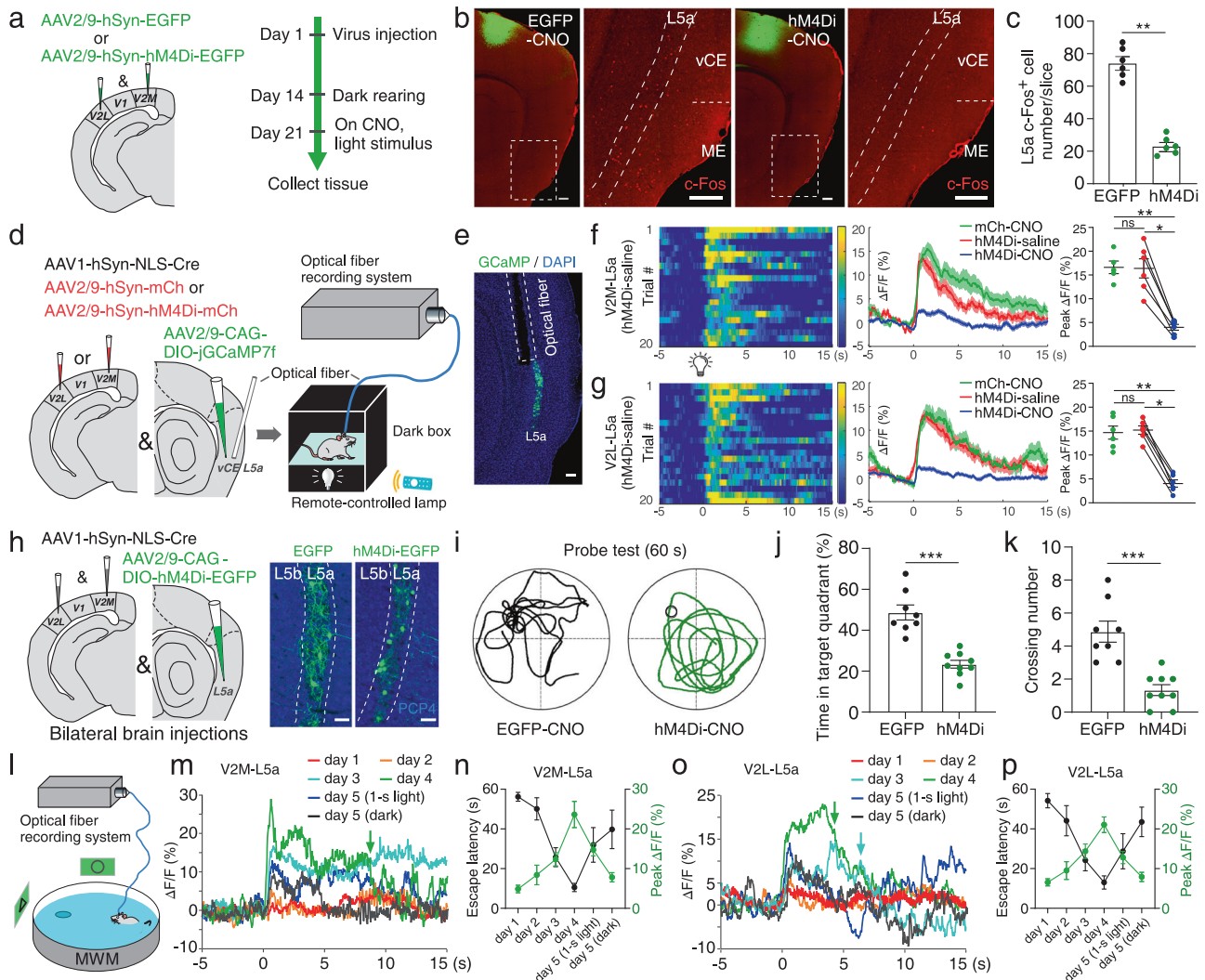

**Fig. 7 | The activity of MEC L5a neurons is involved in spatial navigation.**
**a** Schematic illustrating viral injection. **b** Confocal images showing viral expression (green) and c-Fos expression (red). **c** Comparison of c-Fos expression within MEC L5a ($n = 6$ mice/group; **$P = 0.0022$). **d** Schematic illustrating viral injection and fiber photometry recording. **e** Representative image showing optical fiber placement and GCaMP7f-positive vCE L5a neurons. **f–g** Left: heatmaps showing Ca²⁺ signals of vCE L5a neurons receiving V2M (**f**) or V2L (**g**) projections evoked by light stimuli from a saline-treated mouse with bilateral V2M/V2L infected with hM4Di. Middle: averaged Ca²⁺ transients of mice in different groups; thick lines indicate mean and shaded areas indicate SEM. Right: summary plot exhibiting significantly reduced Ca²⁺ signals caused by inactivation of V2M or V2L neurons (V2M-mcherry, $n = 5$ mice; V2L-mcherry, $n = 6$ mice; V2M-hM4Di, $n = 6$ mice; V2L-hM4Di, $n = 6$ mice). *$P = 0.0313$ (**f** and **g**), **$P = 0.0043$ (**f**), **$P = 0.0022$ (**g**), ns $P ≥ 0.05$. **h**, Left:

schematic illustrating viral injection. Right: representative images showing viral expression within vCE L5a neurons. **i**, Representative swim paths for control and hM4Di mice in MWM tests. **j–k** Summary plot exhibiting time ratio in the target quadrant (**j**) and crossing number over the platform-site (**k**) during the probe test (control, $n = 8$ mice; hM4Di, $n = 9$ mice). ***$P < 10^{-4}$ (**j**), ***$P = 0.0002$ (**k**). **l** Schematic illustrating Ca²⁺ signal recording during MWM tests. **m, o** Representative Ca²⁺ signals of vCE L5a neurons receiving V2M (**m**) or V2L (**o**) projections on different days. Colored arrows indicate the timepoint when the mouse reached the platform on the trace of the same color. **n, p** The escape latencies (black) and their corresponding amplitudes of Ca²⁺ signals (green) on different days ($n = 5$ mice/group). Scale bars, 200 μm (**b**), 100 μm (**e** and **h**). Two-sided Wilcoxon signed-rank test (**f** and **g**, hM4Di-saline vs. hM4Di-CNO), two-sided Mann-Whitney $U$ test (others). Error bars represent SEM. Source data are provided as a Source Data file.

water. Compared with the control group expressing EGFP in V2, light stimulus-induced c-Fos expression in MEC L5a was decreased significantly in the group with V2 inactivated by hM4Di (control group: 74.3 ± 3.9 cells; inactivation group: 23.3 ± 2.2 cells; $P < 0.001$; Fig. 7b, c). These results indicate that neural activity of MEC L5a induced by light stimuli substantially depends on V2.

To further verify the role of the V2→MEC L5a pathway in delivering visual inputs to the MEC, we next used fiber photometry to directly record neuronal responses to the visual stimuli from the vCE L5a neurons receiving V2M or V2L projections (Fig. 7d). Following injection of a virus cocktail of the anterograde trans-synaptic AAV1-hSyn-NLS-Cre and AAV2/9-hSyn-mcherry or AAV2/9-hSyn-hM4Di-mcherry into V2M or V2L, and Cre-dependent virus encoding the

calcium indicator GCaMP7f into vCE L5a of WT mice, GCaMP7f was specifically expressed in the vCE L5a neurons (Fig. 7e). We implanted a small optical fiber (200 μm diameter) with its tip into vCE L5a for chronic recordings of GCaMP fluorescence changes (Fig. 7e). In a dark box, mice were exposed to 2-s light stimuli delivered by a remote-controlled lamp when the spontaneous neuronal activity was low (Fig. 7d). With light turned on, the Ca²⁺ signals of the vCE L5a neurons receiving V2M or V2L projections both increased significantly, whereas chemogenetic inactivation of either V2M or V2L neurons significantly decreased the vCE L5a neuronal response to light stimuli (Fig. 7f, g and Supplementary Fig. 9b, c). By contrast, mice with the optical fiber tip missing vCE L5a did not exhibit light stimulus-evoked signals (Supplementary Fig. 9d). These results

together demonstrate that V2M and V2L both deliver visual information to vCE L5a neurons.

Considering MEC L5a neurons route visual information to superficial MEC and hippocampal CA1, we conjectured that these neurons were involved in spatial navigation. Therefore, we bilaterally delivered anterograde trans-synaptic AAV1-hSyn-NLS-Cre into V2L and V2M, and bilaterally infected MEC L5a neurons with a Cre-dependent virus encoding hM4Di-EGFP or EGFP (Fig. 7h). During training sessions (days 1-6) in MWM, there was no difference in learning ability between the two groups of mice (Supplementary Fig. 10a). During the probe test (day 7) in MWM, sustained suppression of activity in MEC L5a neurons receiving V2 projections was achieved via i.p. injection of CNO (1 mg/kg) in mice of the hM4Di group. The suppression was validated post hoc by in vitro recordings (Supplementary Fig. 10b–e). Compared with control mice of the EGFP group, mice subjected to targeted inactivation of MEC L5a neurons displayed significantly lower spatial memory ability in the probe tests (Fig. 7i–k). The mice in the EGFP group spent significantly more time in exploring the quadrant that previously contained the platform (the target quadrant; 48.7 ± 3.7%), whereas the mice in the hM4Di group explored the target quadrant by chance (23.4 ± 1.9%; $P < 0.0001$; Fig. 7j). The mice in the EGFP group crossed over the platform-site more times than the mice in the hM4Di group (number of times: 4.9 ± 0.6 vs. 1.3 ± 0.3; $P < 0.001$; Fig. 7k). These results indicate that the activity of MEC L5a neurons receiving V2 projections is required in the MWM task, suggesting a critical role of these MEC L5a neurons in spatial memory retrieval.

To further investigate the activity of vCE L5a neurons receiving V2M or V2L projections during spatial navigation, we then used fiber photometry to record $Ca^{2+}$ signals in freely moving mice during the MWM task (Fig. 7l). After several days of training, the mice were proficient in finding the hidden platform quickly on day 4 (Fig. 7n, p). Interestingly, the activity of vCE L5a neurons receiving V2M or V2L projections showed enhancement associated with navigation experience. During the initial exploratory navigation on day 1, an increase in $Ca^{2+}$ activity was barely detectable (Fig. 7m–p). In contrast, on day 4 vCE L5a neurons receiving V2M or V2L projections produced a large and sustained $Ca^{2+}$ signal (Fig. 7m–p). Notably, on day 4 the $Ca^{2+}$ signals rose at the beginning of the navigation and decayed when the hidden platform was reached (Supplementary Fig. 11a, b), indicating that vCE L5a neurons receiving V2M or V2L projections were involved in the navigation task. Next, to determine whether the activity of vCE L5a neurons involved in the navigation task requires visual inputs, on day 5 we gave mice 1-s lighting during the initial phase of navigation and then turned off the ambient light or turned off the ambient light throughout the navigation. In the dark condition, the mice had difficulty in navigating to the platform and the activity of vCE L5a neurons receiving V2M or V2L projections was significantly decreased (Fig. 7m–p; Supplementary Fig. 11a–d). In the 1-s lighting condition, access to visual cues during the initial phase only mildly improved the performance of mice and the activity of the vCE L5a neurons receiving V2M or V2L projections declined after the light was turned off despite its normal initial rising phase (Fig. 7m–p; Supplementary Fig. 11a–d). These results indicate that visual inputs are necessary for the activity of vCE L5a neurons involved in the navigation during the MWM task.

The foregoing results corroborate the critical role of MEC L5a neurons receiving V2 projections during navigation. However, since MEC L5a also receives inputs from other brain areas, the above observations may be attributed to alternative mechanisms, such as learning-related increase in coordination between the deep MEC and the hippocampus[45]. Thus, we would like to investigate features of V2→MEC L5a projections during navigation tasks directly. We first used an optogenetic approach to inquire into the involvement of V2→MEC L5a pathway in spatial navigation. To this end, we bilaterally expressed the inhibitory opsin eNpHR3.0 in V2L or V2M and bilaterally implanted optical fibers above the corresponding V2L or V2M projection sites in

the vCE L5b or L5a (Fig. 8a, b and Supplementary Fig. 12a). During training sessions in MWM and BM (Fig. 8a, e), there was no difference in learning ability between the sham control and eNpHR3.0 groups of mice (Supplementary Fig. 12b and 12c). However, when applying the yellow laser to well-trained mice, optogenetic suppression of either V2L or V2M afferent fibers in the MEC disrupted their ability to directly navigate to the target (Figs. 8c, d and 8f, g). In the sham control group, mice that expressed EGFP in V2L or V2M showed normal escape latencies during the laser was on (Figs. 8c, d and 8f, g). These results demonstrate that both V2L and V2M inputs to the MEC are necessary for proper spatial navigation.

Next, to directly monitor visual inputs to the MEC during navigation, we utilized fiber photometry to record the activity of V2 axons within the MEC. After injecting AAV2/9-CaMKIIα-GCaMP6m into either V2M or V2L and implanting optical fiber above the vCE (Fig. 8h, i and Supplementary Fig. 12d), we first ensured that robust $Ca^{2+}$ responses could be recorded from the V2 axons by applying a brief 2-s light stimulus to freely moving mice (Fig. 8j, k). We then recorded $Ca^{2+}$ signals from V2 axons in mice during the BM task (Fig. 8l). As the training progressed for the first 4 days, the $Ca^{2+}$ signals from both V2M and V2L axons increased in tandem with the marked improvement in locating the escape hole swiftly (Fig. 8m–p and Supplementary Fig. 12e, f). Furthermore, in darkness on day 5, when the activity of V2M and V2L axons decreased, the animals' ability to efficiently locate the escape hole also declined (Fig. 8m–p and Supplementary Fig. 12e, f). These results indicate that sustained visual inputs delivered via the V2→MEC L5a pathway are indispensable for navigation tasks. To summarize, the behavioral data so far together reveal a critical role of the V2→MEC L5a pathway in memory-guided spatial navigation.

## Discussion

### A non-canonical visual cortical-entorhinal pathway

Visual input to the entorhinal cortex (EC) is critical for fundamental brain functions. As a primary sensory source, it not only provides external reference frame and self-motion information for spatial navigation[21,46], but also participates in encoding and retrieving of episodic memory[47]. Previous studies have showed that visual stimuli evoke activities in EC neurons of monkeys and humans[48–50]. However, except for the olfactory input, no direct projection from other unimodal sensory areas to the EC has been reported in the primate[51]. Accordingly, visual input is assumed to enter the EC via polymodal association cortices, especially the perirhinal cortex (PER) and POR projecting to superficial layers of LEC and MEC, respectively[51]. However, this dogma has been challenged by a recent study showing that both PER and POR mainly target the rat LEC, with only minimal projections from POR to MEC[31]. Thus, the role of the MEC in sensory information integration and relay has become more elusive.

In the present study, by monitoring the c-Fos expression in dark reared mice[29,30], we corroborated that neurons in the MEC responded to brief visual stimuli. However, to our surprise, c-Fos expressing neurons evoked by visual stimulus were dominated by L5a neurons, but not superficial neurons in the MEC. In the ensuing experiments, employing anterograde trans-multisynaptic/-monosynaptic and retrograde trans-monosynaptic viral tracers, we revealed a previously unidentified visual cortical-entorhinal pathway, i.e., V2→MEC L5a pathway (Fig. 9). Subsequent in vitro optogenetic experiments demonstrated that this pathway was mediated by functional synapses. The observation that L5a neurons responded to visual stimuli inspired our finding of V2→MEC L5a pathway, but does this pathway really route visual input into the MEC? To address this issue, we silenced the activity of V2 using the DREADD-based chemogenetic tool and then found an impairment in the visual-stimulus-evoked c-Fos expression of MEC L5a neurons. Moreover, fiber photometry recordings of calcium signals demonstrated real-time responses of MEC L5a neurons to visual

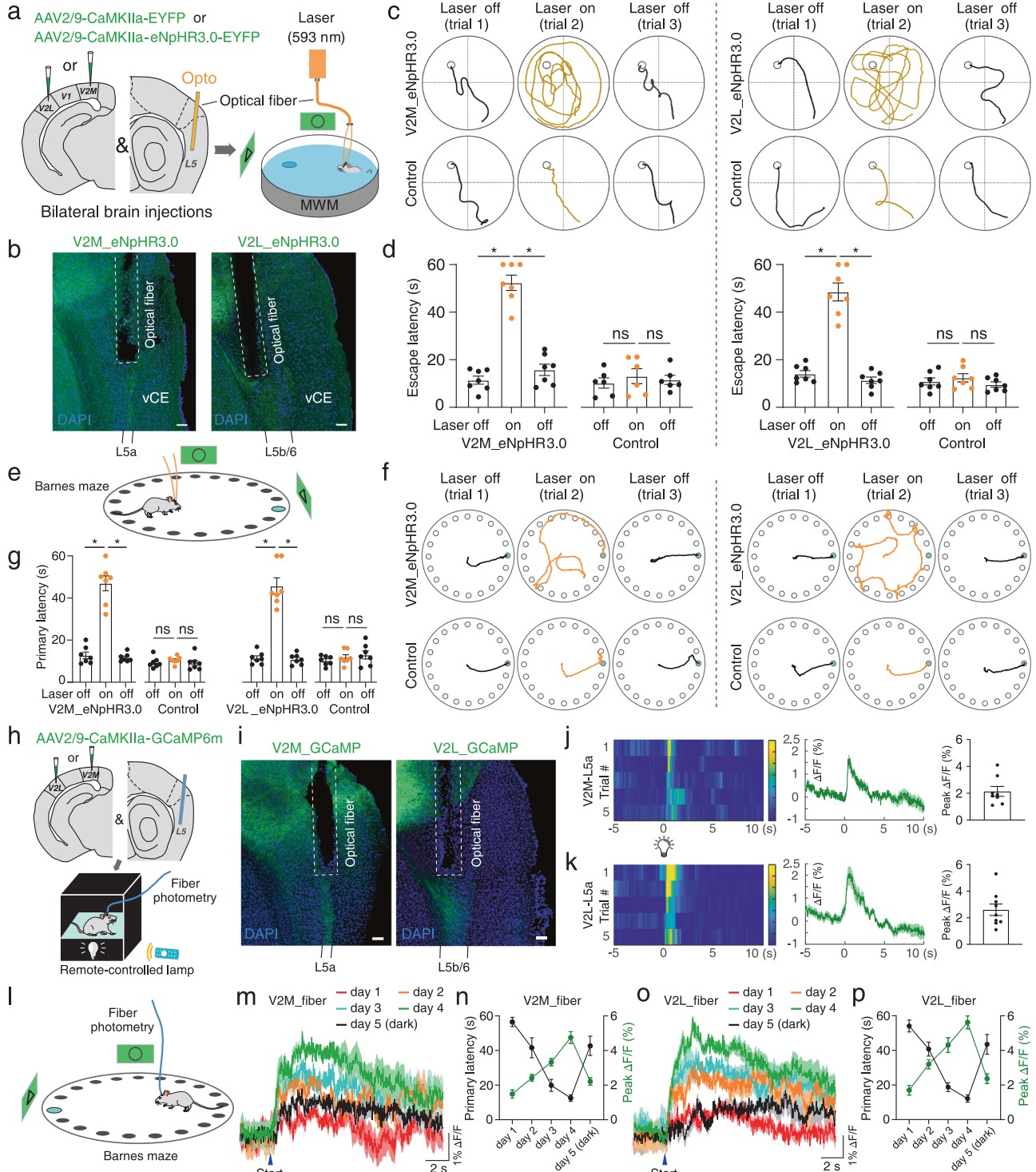

**Fig. 8 | The V2→MEC L5a pathway is critical for spatial navigation. a** Schematic illustrating viral injection and optogenetic manipulation during MWM tests. **b** Representative image showing optical fiber placement and eNpHR3.0-expressing fibers. **c** Representative swim paths for well-trained mice with virus injected in V2M (left) or V2L (right) during consecutive light off-on-off trials. **d** Summary plot exhibiting the effect of light-induced eNpHR3.0 activation on escape latency (V2M-eNpHR3.0, *n* = 7 mice; V2M-EGFP, *n* = 6 mice; V2L-eNpHR3.0, *n* = 7 mice; V2L-EGFP *n* = 7 mice). *P = 0.0156, ns *P* ≥ 0.05. **e** Schematic illustrating optogenetic manipulation during BM tests. **f** Representative movement trajectory for well-trained mice with virus injected in V2M (left) or V2L (right) during consecutive light off-on-off trials. **g** Summary plot exhibiting the effect of light-induced eNpHR3.0 activation on primary latency (V2M-eNpHR3.0, *n* = 7 mice; V2M-EGFP, *n* = 7 mice; V2L-eNpHR3.0, *n* = 7 mice; V2L-EGFP *n* = 7 mice). *P = 0.0156, ns *P* ≥ 0.05.

**h** Schematic illustrating viral injection and fiber photometry recording. **i** Representative image showing optical fiber placement and GCaMP-expressing fibers. **j–k** Left: heatmaps showing the Ca$^{2+}$ signals of V2M fibers (**j**) or V2L fibers (**k**) evoked by light stimuli. Middle: averaged Ca$^{2+}$ transients of mice in different groups; thick lines indicate mean and shaded areas indicate SEM. Right: summary plot exhibiting the peak of Ca$^{2+}$ signals of V2M (**j**, *n* = 8 mice) or V2L (**k**, *n* = 9 mice) fibers. **l** Schematic illustrating fiber photometry recording during BM tests. **m**, **o** Mean Ca$^{2+}$ signals of V2M (**m**, *n* = 8 mice) or V2L (**o**, *n* = 9 mice) fibers on different days. Shaded areas indicate SEM. **n**, **p** The escape latencies (black) and their corresponding amplitudes of Ca$^{2+}$ signals (green) on different days (**n**, V2M, *n* = 8 mice; **p**, V2L, *n* = 9 mice). Scale bars, 100 μm (**b** and **i**). Two-sided Wilcoxon signed-rank test. Error bars represent SEM. Source data are provided as a Source Data file.

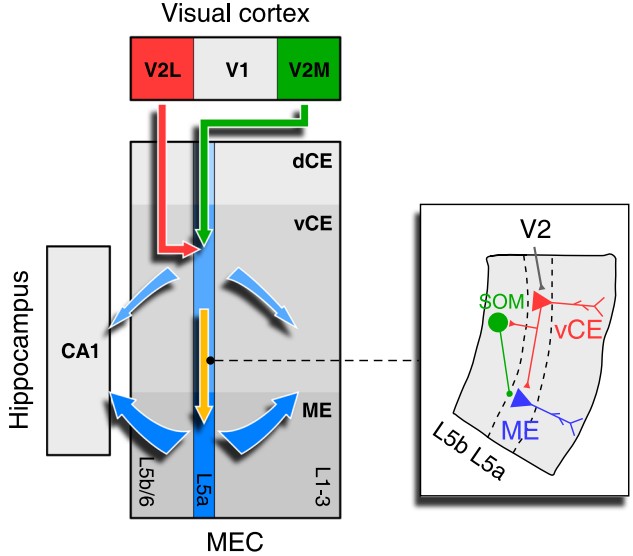

**Fig. 9 | Schematic summarizing the visual cortical-entorhinal pathway revealed in the present study.** Neurons in vCE L5a receive visual inputs via direct synaptic connections from V2M and V2L, and further route them to MEC superficial layers and hippocampal CA1 either directly or indirectly via ME L5a, with the latter pathway dominating. Also indicated are SOM INs in deep layers, which regulate the DV pathway, i.e., vCE L5a to ME L5a, via feedforward inhibition.

stimuli and these responses were also impaired by silencing the activity of V2. Thus, our results causally demonstrate a non-canonical pathway from V2 directly to MEC L5a, gating visual information transmission to the MEC.

## L5a neurons as a network hub for visual processing in the MEC

The V2 consists of the medial and lateral subdivisions, i.e., V2M and V2L. Based on anatomical and functional features, previous studies have further identified multiple HVAs, including the POR[38,39,52]. Of note, V2M has been observed to project to deep layers of the MEC[26], and the minimal projections from POR to MEC mentioned above are also distributed in deep layers of the CE[31] and form synaptic connections with L5 PCs[53]. The present study confirms and further expands these findings. Our anterograde and retrograde tracing experiments showed that both V2M and V2L target vCE L5a neurons and the presynaptic neurons were distributed in multiple HVAs, suggesting that L5a is essentially an input hub receiving convergent visual inputs from various regions of the V2 (Fig. 9). On the other hand, L5a neurons project to a lot of other telencephalic structures[32,33], as well as key regions for spatial navigation, such as MEC L2/3[33] and hippocampal CA1[23,24]. Our tracing and optogenetic experiments further demonstrated that vCE L5a neurons receiving V2 inputs formed functional synapses with MEC L2/3 and CA1 neurons. Thus, L5a also functions as an output hub routing visual information divergently to multiple areas (Fig. 9).

Intriguingly, as a visual processing hub, L5a is not a simple node, but is configured in the form of a feedforward network. Our data showed that the direct relay of visual inputs to L2/3 or CA1 by vCE L5a was marginal. Instead, vCE L5a output to L2/3 and CA1 was mainly relayed by ME L5a, thus suggesting an information stream along the DV axis within L5a (Fig. 9). This finding is consistent with an early study showing that in the rat MEC, L5a as well as other layers of the midtemporal part (i.e., vCE) all have longitudinal projections directed primarily toward the ventral part[54]. A separation of the input and output layers in a local neural network is ubiquitous throughout the cortex, but they are usually organized according to cortical layers. For example, hippocampal outputs to the telencephalic structures are relayed by MEC L5 in a manner of multi-stage processing, with L5b and L5a largely engaged in receiving and sending hippocampal outputs,

respectively[25]. However, in the opposite direction, the present study shows that the multi-stage processing is performed by neuron populations at different longitudinal positions, i.e., vCE L5a and ME L5a. Nonetheless, it is notable that the local processing network still involves L5b, but mainly INs therein. Our data showed that in mice subjected to visual stimulus, L5b harbored a relatively small number of c-Fos positive neurons and more than half of them were INs. Furthermore, the tracing experiments displayed that vCE L5a neurons receiving the V2M/V2L input innervated INs of vCE L5b overwhelmingly, in agreement with the sparse connections from L5a to L5b excitatory neurons observed in dual whole-cell recordings[55]. In particular, SOM INs dominated the vCE L5b INs targeted by vCE L5a and engaged in feedforward inhibition from vCE L5a to ME L5a. SOM INs of the MEC as a whole have been shown to maintain selectivity of aperiodic spatial cells[56] and modulate working-memory formation[57], but the exact function of SOM INs in L5b is unknown. Our data suggest that SOM INs in L5b may influence the visual information transmission through the vCE-ME pathway by maintaining the balance of excitation and inhibition, sharpening the temporal precision of neural responses, and/or modulating gain and dynamic range of neural activity[58].

## Functional implications for V2→MEC L5a pathway

As the major gateway of hippocampal outputs to the neocortex, MEC L5a has been reported to support long-term memory formation via projections to the prefrontal cortex[59] and regulate depression-like behaviors via projections to V2M[41]. However, the function of projections in the opposite direction was rarely studied. In the present study, with chemogenetic silencing of L5a neurons targeted by V2, the performance of animals in MWM tests was impaired, indicating an inability to retrieve spatial memory. EC lesions have been reported to impair spatial navigation in MWM[60–62]. Here we show that V2-targeted neurons in lateral vCE L5a are necessary for intact spatial memory retrieval. To further isolate the effect of V2→MEC L5a pathway, the projections from V2M or V2L to MEC were directly inhibited by optogenetic activation of eNpHR3.0. Our results of MWM and BM demonstrated that both V2M and V2L inputs were indispensable for spatial navigation and the associated neural activity of MEC L5a. In the primate visual system, the cortical network has been divided into a dorsal "where" stream involved in visual guidance and a ventral "what" stream involved in object recognition[63]. In the rodent visual cortex, previous studies have also provided evidence for the existence of multiple parallel processing streams originating from HVAs in V2[64,65]. The prevalent view assumes that the dorsal "where" and the ventral "what" streams are mediated by MEC and LEC, respectively, but this model has been recently challenged by Doan et al., 2019[31] as already mentioned. A three-color tracing study in mice also indicated that both dorsal and ventral streams input to LEC[66]. Furthermore, in striking contrast to the prevalent model, the study showed that the input to MEC is dominated by the ventral "what" but not the dorsal "where" stream[66], revising the result of an earlier paper from the same lab[67]. In addition, the difference between responses of grid cells in the dorsal and the ventral subdivisions of MEC to changes of the environmental barrier[68] has inspired the speculation that these MEC subdivisions belong to different streams[16,64]. Thus, there are divergent opinions concerning the organization of parallel streams in the EC. The observation in the present study further reveals the complexity of the issue. Although V2M and V2L both targeted MEC L5a and were both required for proper spatial navigation, their projecting fibers had staggered distributions and preferred to innervate different populations of L5a neurons. Further studies will be necessary to determine whether V2M and V2L inputs to MEC L5a deliver different visual information.

Besides, V2→MEC L5a pathway may also provide external reference for grid cells. As mentioned above, there are multiple HVAs surrounding V1[38,39,52]. These HVAs have different preferences for representing particular portions of the visual field in the mouse[52]. RL

responds to the ventral visual field, whereas LM and P respond to the dorsal visual field, with other HVAs less biased. In the present study, we demonstrated that vCE L5a was targeted by multiple HVAs except for RL/A. Meanwhile, V2 rarely targeted dCE L5a. Therefore, vCE L5a, but not dCE L5a, receives more information from the dorsal visual field. In a modeling study[16], the dorsal visual field has been assumed to provide information of distal landmarks to the ventral MEC for explaining the observation that the firing fields of grid cells in more ventral regions exhibited compressed spacing when the environmental barriers were compressed, but those in more dorsal regions could retain their spacing[68]. These findings together imply a potential role of V2→vCE L5a pathway in the modulation of grid cell firing fields by visual cues. Indeed, L5 neurons have special advantages for the task. First, L5 has the largest proportion of conjunctive cells. 90% of L5 grid cells have conjunctive properties[2]. In continuous attractor models, conjunctive cells have been assumed to drive the shift of the firing fields of grid cells[69]. V2→vCE L5a pathway enables the visual input to recruit vCE L5a neurons to influence the firing field locations. Second, L5 neurons display significant persistent firing at graded firing rates[70], which is believed to be an elementary mechanism for working memory of static visual features and therefore make animals capable of using previously captured non-real-time visual cues for computations[16]. Persistent firing has been observed in neurons across layers of the MEC[70–72], but in contrast to superficial neurons, L5 neurons strengthen their persistent firing with repetitive excitatory input, suggesting a preference for coding familiar information. Third, L5a neurons project to both superficial layers of MEC and hippocampal CA1[23,24,33]. Both MEC grid cells and CA1 place cells anchor their firing fields to visual cues[1,10–12,68]. Thus, V2→vCE L5a pathway may play a role in coordinating the modulation of visual input on the neural activity in both regions.

In addition to spatial navigation, V2→MEC L5a pathway could also engage in fear- and anxiety-related behaviors. Our data demonstrated that via vCE and ME L5a, visual inputs could be delivered to vCA1, which has dense connectivity with the amygdala as well as the hypothalamus and has been linked to fear, anxiety, stress and defensive behaviors[73,74]. In particular, the ventral hippocampus contains place cells with a much larger spatial scale, which has been assumed to endow animals with the advantage of detecting faraway approaching danger[73]. A shortcut for visual signals to reach vCA1 across fewer synapses should further benefit animals in making an early response to visible danger like predators. V2→MEC L5a pathway is probably competent to carry out the task. Additionally, MEC L5a also projects to the amygdala directly[32,33]. Therefore, it will be of great interest for future investigations to verify the role of this pathway in fear- and anxiety-related behaviors.

## Methods

### Experimental animals
Adult male and female mice (2–4 months) were used for all experiments. The Cre and Ai9 reporter lines were purchased from Jackson Laboratory (Vgat[Cre], #028862; SOM[Cre], #013044; PV[Cre], #017320; Ai9, #007909). The Vgat[Cre] × Ai9, SOM[Cre] × Ai9 and PV[Cre] × Ai9 mice were generated by crossing the corresponding Cre line with the Ai9 line. The mice were housed under standardized conditions with a 12 h light/dark cycle, with temperature controlled at 22–25 °C and humidity at 40–60%. All procedures for animal surgery and maintenance were performed following protocols approved by the Institutional Animal Care and Use Committee (IACUC) of Harbin Institute of Technology.

### Dark rearing and brief light stimuli
In dark rearing experiments, 3-month-old mice were fed in dark chambers for 7 days. After this period, the light was turned on and the mice were exposed to the brief light stimulus for 2 min, and then the light was turned off. 1 h later, the mice were deeply anesthetized by inhaled isoflurane for immunohistochemistry.

### Viral constructs and injection strategy
The injection pipette was at a 18° angle towards the caudal end of the brain in all surgeries. We set the position of bregma as $X = 0$ mm, the transverse sinus position as $Y = 0$ mm, the surface of the brain as $Z = 0$ in all surgeries. For targeting vCE L5a of MEC, a craniotomy was made at $X = 3.5$–$3.75$ mm, $Y = 0.7$–$0.8$ mm, and virus was slowly released at $Z = 2.6$–$2.8$ mm. For targeting other regions, the coordinates were as follows: ME L5a ($X = 3.5$–$3.75$ mm, $Y = 0.7$–$0.8$ mm, $Z = 3.2$–$3.4$ mm), MEC L2 ($X = 3.5$–$3.75$ mm, $Y = 0.4$–$0.5$ mm, $Z = 1.8$–$3.0$ mm), V2L ($X = 3.5$–$3.6$ mm, $Y = 2.0$–$2.2$ mm, $Z = 0.7$–$1.0$ mm), V2M ($X = 1.3$–$1.5$ mm, $Y = 1.9$–$2.0$ mm, $Z = 0.7$–$0.9$ mm), V1 ($X = 2.3$–$2.5$ mm, $Y = 1.7$–$1.9$ mm, $Z = 0.6$–$0.8$ mm), iCA1 ($X = 3.4$–$3.6$ mm, $Y = 2.0$–$2.2$ mm, $Z = 2.1$–$2.2$ mm) and vCA1 ($X = 3.4$–$3.6$ mm, $Y = 2.0$–$2.2$ mm, $Z = 3.4$–$3.5$ mm). After delivering the virus, the injection pipette was left in place for an additional 10 min before it was fully withdrawn.

Viruses used in this study were as follows: AAV2/9-EF1α-TK-tdTomato (titer: $3.2 \times 10^{12}$ v.g./ml; 100 nl/injection) and H129-ΔTK-EGFP (titer: $5.0 \times 10^{8}$ PFU/ml; 60 nl/injection) for trans-monosynaptic tracing strategy; H129-hUbC-HBEGFP (titer: $1.0 \times 10^{9}$ PFU/ml; 40 nl/injection) for trans-multisynaptic tracing strategy; AAV2/9-hSyn-EGFP (titer: $5.14 \times 10^{12}$ v.g./ml; 100 nl/injection) and AAV2/9-hSyn-mCherry (titer: $5.0 \times 10^{12}$ v.g./ml; 100 nl/injection) for visualizing V2 axons in MEC; AAV2retro-CAG-EGFP (titer: $5.0 \times 10^{12}$ v.g./ml; 100 nl/injection) and AAV2retro-CAG-mCherry (titer: $5.0 \times 10^{12}$ v.g./ml; 100 nl/injection) for retrograde tracing from V2M and V2L; AAV2retro-hSyn-Cre (titer: $5.14 \times 10^{12}$ v.g./ml; 150 nl/injection), cocktail of AAV2/9-EF1α-DIO-EGFP-T2A-TVA and AAV2/9-EF1α-DIO-oRVG (titer: $1.2 \times 10^{12}$ v.g./ml; 150 nl/injection), RV-CVS-ENVA-N2C(ΔG)-tdTomato (titer: $3.0 \times 10^{8}$ TFU/ml; 70 nl/injection) for retrograde tracing from MEC L5a; AAV2/9-CaMKII-hChR2(H134R)-mCherry (titer: $5.60 \times 10^{12}$ v.g./ml; 100 nl/injection) for optically activating V2 projections to MEC L5a; AAV2/9-SOM-EGFP (titer: $5.11 \times 10^{12}$ v.g./ml; 30 nl/injection) and AAV2/9-SOM-eNpHR3.0 (titer: $5.53 \times 10^{11}$ v.g./ml; 30 nl/injection) for labeling and optically inhibiting SOM-interneurons of MEC; portfolio strategy of AAV2/9-CaMKII-DIO-hChR2(H134R)-mCherry (titer: $2.66 \times 10^{12}$ v.g./ml; 100 nl/injection) and AAV1-hSyn-SV40 NLS-Cre (titer: $1.08 \times 10^{13}$ v.g./ml; 150 nl/injection) for activating MEC L5a pyramidal neurons; cocktail of AAV2/9-hSyn-DIO-EGFP (titer: $5.08 \times 10^{12}$ v.g./ml; 100 nl/injection) and AAV2retro-hSyn-Cre (titer: $5.14 \times 10^{12}$ v.g./ml; 150 nl/injection) for labeling ME L5a neurons. AAV2/9-hSyn-EGFP (titer: $5.14 \times 10^{12}$ v.g./ml; 90 nl/injection) and AAV2/9-hSyn-hM4Di-EGFP (titer: $4.90 \times 10^{12}$ v.g./ml; 90 nl/injection) for the DREADD-based inhibition of V2; AAV2/9-CaMKIIα-GCaMP6m (titer: $5.21 \times 10^{12}$ v.g./ml; 100 nl/injection), AAV2/9-CAG-DIO-jGCaMP7f (titer: $4.65 \times 10^{12}$ v.g./ml; 100 nl/injection) and cocktail (200 nl/injection) of AAV1-hSyn-SV40 NLS-Cre (titer: $1.10 \times 10^{12}$ v.g./ml) and AAV2/9-hSyn-mCherry (titer: $5.50 \times 10^{11}$ v.g./ml) or AAV2/9-hSyn-hM4Di-mCherry (titer: $6.50 \times 10^{11}$ v.g./ml) for fiber photometry recordings of calcium signal; AAV2/9-hSyn-DIO-EGFP (titer: $5.08 \times 10^{12}$ v.g./ml; 100 nl/injection), AAV2/9-hSyn-DIO-hM4Di-EGFP (titer: $2.95 \times 10^{12}$ v.g./ml; 120 nl/injection) and AAV1-hSyn-SV40 NLS-Cre (titer: $1.08 \times 10^{13}$ v.g./ml; 150 nl/injection) for the DREADD-based inhibition of vCE L5a; AAV2/9-CaMKII-eNpHR3.0-EYFP (titer: $4.68 \times 10^{12}$ v.g./ml; 100 nl/injection) or AAV2/9-CaMKII-EYFP (titer: $5.44 \times 10^{12}$ v.g./ml; 100 nl/injection) for optogenetic inhibition of projecting axons from V2. For HSV and RV tracing experiments, all procedures on animals were performed in biosafety level 2 (BSL-2) animal facilities. All viruses and dye were purchased from BrainVTA (Wuhan, China), Brain Case Biotech (Shenzhen, China) and OBiO Technology (Shanghai, China).

### Immunofluorescence
Deeply anesthetized mice were transcardially perfused with ice-cold saline and then followed by 4% cold paraformaldehyde (PFA) in 0.1 M phosphate buffer (PB). After overnight fixation in cold PFA, brains and immersion-fixed acute slices were sectioned into 40 μm-thin sections

with a vibratome (Leica, VT1200S). The method of tangential section followed a previous study[75]. On the selected sections, the following primary antibodies were used: mouse anti-c-Fos IgG (1:500; Abcam, ab208942); rat anti-Ctip2 IgG (1:500; Abcam, ab18465); rabbit anti-GAD65/67 IgG (1:500; Abcam, ab11070); rabbit anti-CaMKII IgG (1:500; Abcam, ab52476); rabbit anti-PCP4 IgG (1:500; Sigma, HPA005792); rat anti-Somatostatin IgG (1:200; Millipore, MAB354). All primary antibodies were diluted in 0.1 M PB and incubated overnight at 4 °C. For fluorescence labeling, the following secondary antibodies were applied on the sections: Alexa-488-conjugated goat anti-mouse IgG (1:1000; Abcam, ab150113); Alexa-488-conjugated donkey anti-rabbit IgG (1:1000; Abcam, ab150073); Alexa-405-conjugated goat anti-rabbit IgG (1:1000; Abcam, ab175652); Alexa-555-conjugated goat anti-rabbit IgG (1:1000; Abcam, ab150078); Alexa-555-conjugated goat anti-mouse IgG (1:1000; Cell signaling technology, #4409); Alexa-647-conjugated goat anti-mouse IgG (1:1000; Cell signaling technology, #4410) and Alexa-405-conjugated donkey anti-rat IgG (1:1000; Abcam, ab175670). Sections were examined and imaged using a laser scanning confocal microscope (Zeiss, LSM 880) controlled by ZEN Black software (Zeiss, version 3.0). ZEN 2 software (Zeiss, version 2.6) and ImageJ software (NIH, version 1.53) were also used to count the labeled neurons.

## Slice preparation
Electrophysiological experiments were performed in acute sagittal or horizontal slices taken from the mice that were previously injected intracranially with AAV. The mice were deeply anesthetized by inhaled isoflurane and decapitated. The brains were quickly removed and placed in ice-cold (0–4 °C) artificial cerebrospinal fluid (ACSF) solution containing 119 mM NaCl, 2.5 mM KCl, 1 mM NaH$_2$PO$_4$, 26 mM NaHCO$_3$, 1 mM MgCl$_2$, 25 mM glucose and 2 mM CaCl$_2$, saturated with carbogen (95% O$_2$ and 5% CO$_2$), pH 7.4. Sagittal or horizontal slices containing MEC and hippocampus (300 μm thick) were cut with a vibratome (Leica, VT1200S), and incubated for 10 min at 34 °C in oxygenated ACSF solution. The slices were then maintained at room temperature for an additional 0.5 h until being transferred to the recording chamber. After electrophysiological recordings, the sections were immersed into fixative (4% PFA in 0.1 M PB) for overnight fixation.

## In vitro electrophysiological recordings
Whole-cell recordings were performed in a similar way as our previous studies[76–78]. Patch recording pipettes (4–7 MΩ) were filled with intracellular solutions containing (mM): 135 cesium methanesulfonate, 10 HEPES, 2.5 MgCl$_2$, 4 Na$_2$ATP, 0.4 Na$_3$GTP, 10 sodium phosphocreatine, 0.6 EGTA, 0.1 spermine and 0.5% biocytin, at pH 7.25 for current recordings; or 120 potassium gluconate, 10 HEPES, 4 KCl, 4 MgATP, 0.3 Na$_3$GTP, 10 sodium phosphocreatine and 0.5% biocytin, at pH 7.25 for voltage recordings. Whole-cell recordings were made with Axopatch 200B patch clamp amplifiers (Molecular Devices). Data were collected with Igor Pro (WaveMetrics, version 6.0). During the recording, the slices were continuously perfused with oxygenated ACSF solution at 30 °C throughout the recording session.

For optogenetic activation of ChR2-expressing neurons, blue light (2.5 mW/mm$^2$, 3 ms duration) from a 470-nm low-noise LED (Prizmatix, UHP-T-470-SR-LN) was delivered to the slices through the excitation light path of a fluorescence microscope (Olympus, BX51WI). For optogenetic inhibition of eNpHR3.0 expressing neurons, yellow light (3 mW/mm$^2$, 1 s duration) from a 593-nm laser of an optogenetic stimulation system (Thinker Tech, QAXK-JGQ) was delivered to the slices by the guidance of an optical fiber. EPSCs and IPSCs were recorded at −70 mV and 0 mV, respectively. TTX (1 μM, Tocris) and 4-AP (200 μM, MedChemExpress) were used to block Na$^+$ and K$^+$ channels, enabling functional assessments of monosynaptic synaptic connectivity. Picrotoxin (100 μM, MedChemExpress), CNQX (10 μM, MedChemExpress) and AP-5 (50 μM, MedChemExpress) were used to block GABA$_A$ receptor, AMPA receptor and NMDA receptor mediated currents, respectively.

For post hoc validation of the DREADD-based inhibition, the hM4Di-EGFP- or EGFP-expressing MEC L5a neurons were recorded in current-clamp mode. Depolarizing step currents (100–400 pA, 1 s) were injected into the neuron to evoke the firing of APs before and 10 min after the application of CNO (5 μM; APExBIO, A3317) in the recording solution. The change in the spike numbers was analyzed to exhibit the effect of hM4Di/CNO.

## Histology and morphological reconstructions
The avidin-biotin-peroxidase method was used to reveal neuronal morphology[76,78]. Briefly, after whole-cell recording, the brain slice was quickly transferred into 4% PFA in 0.1 M PB and fixed for at least 48 h at 4 °C. The fixed slices were then removed from PFA solution and rinsed in PBS. To restrain endogenous peroxidase activity, slices were quenched in 3% H$_2$O$_2$ for 30 min. Following renewed rinses in PBS, slices were permeabilized with 2% Triton X-100 in PBS (Triton-PB) for 30 min to prevent nonspecific binding. Subsequently, the brain slices were incubated with an avidin-biotin-horseradish peroxidase complex (Vector Laboratories) for 3 h. After thorough rinses in PBS, the brain slices were developed with 3,3-diaminobenzidine solution (DAB substrate solution). When the desired staining intensity was achieved, the DAB reaction was stopped by rinsing in 0.1 M PB, and then the brain slices were mounted on glass slides and embedded in Mowiol. After Mowiol was dried, the morphologically recovered cells were examined and reconstructed using the Neurolucida (MBF Bioscience, version 11) on an upright microscope (BX53, Olympus).

## Fiber photometry
Following the injection of AAV2/9-DIO-GCaMP7f, an optical fiber (200 μm O.D., NA 0.37; Thinker Tech) was placed in a ceramic ferrule (2.5 mm O.D.) and inserted towards vCE L5a through the craniotomy. Ceramic ferrule was supported with superglue and dental acrylic. Mice were individually reared for at least 2 weeks to recover. Neuronal Ca$^{2+}$ signals were acquired at 40 Hz with a three-color multichannel optical fiber recording system (Thinker Tech, QAXK-FPS-SS-MC-LED) controlled by Tripple Color Multi-Channel Fiberphotometry software (Thinker Tech, version 2.0). The green channel (499–529 nm) with 470-nm excitation light was used to record the fluorescence of GCaMP7f. An optical fiber guided the light between the recording system and the implanted optical fiber. The light power at the tip of the optical fiber was adjusted to a low level of 0.01–0.02 mW in order to reduce bleaching.

To verify L5a neurons respond to light stimuli, on testing days, mice were fitted with an optical fiber and placed in a dark box. A remote-controlled lamp was placed below the mice to give light stimulation without interfering fluorescence signals. Before recording, mice in the mCherry or hM4Di-mCherry groups received an i.p. injection of CNO (1 mg/kg; APExBIO, A3317) or saline and were free to move or eat in the dark box for 30 min to improve sensitivity to light stimuli. During recording, the mice were given a 2-second light stimulus delivered by the remote-controlled lamp. To verify L5a neurons were involved in the MWM task, mice were fitted with an optical fiber to swim in the pool. During each trial, mice were allowed to search the platform for 60 s. An infrared camera was used so that recordings could be made even in darkness. All the fluorescence data were segmented and aligned to the onset of light stimuli or MWM task within individual trials. The fluorescence difference relative to the baseline (ΔF/F) was calculated to quantify the neural response to the light stimuli.

## Morris water maze
Spatial learning and memory were assessed with the hidden platform version of the Morris water maze test. The test apparatus consisted of a

circular pool (122 cm diameter, 70 cm depth) filled with water ($24 \pm 2\,°C$) to a depth of 40 cm. The water was made opaque with titanium dioxide to prevent the mice from seeing the platform (10 cm diameter) submerged 1 cm beneath the water surface. The pool was divided into 4 virtual quadrants and the platform was submerged at a fixed spatial position in one of the quadrants. Four distal cues were placed on the pool wall to help spatial navigation. The task comprised a 6-day spatial acquisition phase (1 training session per day, containing 3 trials with different start positions in three quadrants other than the target quadrant, respectively) and a 1-day probe trial. Mice were placed in the water facing the tank wall and allowed 60 s to find and mount the platform. After finding the platform, mice were allowed to rest on the platform for 30 s and were then placed in a holding cage. The animals that failed to find the platform in 60 s were guided to the platform. 24 h after the last training day, the probe trial was performed. The probe trial consisted of a 60 s free swim period without the platform. Time spent in the target quadrant and number of crossings over the former platform location were recorded with Smart 3.0 video tracking software (Panlab Harvard Apparatus). During the probe trial, sustained suppression of activity in MEC L5a neurons receiving V2 projections was achieved via i.p. injection of CNO (1 mg/kg; APExBIO, A3317) in the hM4Di group of mice.

## Barnes maze
Mice were also tested for spatial cognition in the Barnes maze (90 cm diameter, 20 holes, 5 cm hole diameter, 60 cm height). Four visual cues in different shapes and colors were placed around the maze. All sessions were recorded using an infrared camera. A bright light was affixed above the maze to motivate the escape behavior. The Barnes maze was cleaned before and after each testing trial with 75% ethanol. During the training phase (days 1–5), mice were tested for 2 trials per day. For each trial, mice were first placed in an opaque cup (for Fig. 8e) or a transparent cup (for Fig. 8l to avoid abrupt light change) in the center of the maze for 5 s. Then, the cup was lifted, and mice were allowed to explore the maze freely for 3 min. After finding the escape box, mice were allowed to rest in the box for 2 min before returning to cages. The animals that failed to find the escape box in 3 min were slowly guided to the target hole and gently pushed into the escape box if they did not enter by themselves. The probe trial was performed on day 6 (for Fig. 8e). The fiber photometry recording was performed on days 1–5 (for Fig. 8l). For each trial, mice were allowed to explore the maze for 1 min. Primary latency was defined as the time to first visit the target hole and was recorded over the trainings.

## Optogenetic inhibition in behaving mice
The implanting of optical fiber (200 μm O.D., NA 0.37) into vCE L5a was performed on both hemispheres following a procedure as described above. The recovered mice received a 6-day spatial acquisition training in Morris water maze or a 5-day spatial acquisition training in Barnes maze as described above. On the testing days, an additional session consisting of 3 trials was conducted. The optogenetic inhibition laser was turned off in the 1st and 3rd trials but turned on in the 2nd trial. Control mice underwent the same procedure and received the same intensity of laser stimulation. Light power used in the optogenetic inhibition was 18 mW.

## Statistics and reproducibility
Statistical results were reported as mean ± SEM, unless otherwise indicated. Statistical significances were determined using Mann-Whitney U and Wilcoxon signed-rank tests as described in the text. Statistical significance was set at $P < 0.05$. Data analysis described above was performed using Igor Pro (WaveMetrics, version 6.0), GNU Octave (version 7.1.0), Python (version 3.8), RStudio (version 1.4.1717) and Microsoft Excel (Microsoft, version 2019). Representative images presented in Figs. 3e, 4a, 4e, 5b, 7e, 7h, 8b and 8i are from independent experiments with similar results (3e, $n = 3$; 4a, $n = 7$; 4e, $n = 4$; 5b, $n = 4$; 7e, $n = 23$; 7h, $n = 8$ for EGFP and $n = 9$ for hM4Di; 8b, $n = 7$ for V2M and $n = 7$ for V2L; 8i, $n = 8$ for V2M and $n = 9$ for V2L).

## Reporting summary
Further information on research design is available in the Nature Portfolio Reporting Summary linked to this article.

## Data availability
All data supporting the results of this study are available within the paper and its Supplementary Information. Raw data are too large to be publicly shared but are available from the corresponding author upon request. Source data are provided with this paper.

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

## Acknowledgements
We thank all members of the Wang laboratory for comments and technical assistance. This work was supported by the National Key R&D Program of China (2022YFA1604502 to G.W.) and the National Natural Science Foundation of China (31970912 to G.W.).

## Author contributions
Qm.S. and G.W. conceived the study. Qm.S. performed the immunohistochemistry and neuroanatomy. Qm.S., L.C., M.L. Q.S. and X.Z. developed in vivo experimental techniques and assisted in the setup maintenance. Qm.S. conducted the surgeries and behavior experiments. G.W. developed whole-cell recording system. Qm.S., Xw.L., Y.S. and K.Y. performed whole-cell recordings and assisted in the maintenance and the operation of the electrophysiological setup. Qm.S., Xw.L., H.C., Xy.L., Y.S. and K.Y. performed the morphological reconstructions and analysis. Qm.S. and L.C. analyzed the data. Qm.S and G.W. wrote the manuscript with the input from all co-authors.

## Competing interests
The authors declare no competing interests.
