## [Peer Review File · Nature Communications]

REVIEWER COMMENTS

Reviewer #1 (Remarks to the Author):

In this study, Shao et al., investigate how visual signals enter the Medial Entorhinal Cortex (MEC). Using a variety of tracing tools, they identify the direct and indirect targets of visual inputs to the MEC and draw very interesting and novel conclusions about the circuitry downstream of the visual inputs. Mainly, they find that projections to MEC primarily originate from the secondary visual area and directly target neurons in L5a which are responsive to light stimulation. Therefore, they conclude that the property of L5a neurons responding to light is due to the signals from V2. They further show that quite interesting V2M and V2L pathways target different subtypes of L5a neurons at different dendritic positions and move on to characterizing the downstream neurons and topographic relationships and synaptic properties of the connections between neurons downstream of the visual inputs. They then perform functional tests to probe the functional contribution of the visual inputs to MEC for spatial navigation.

The study is extremely timely and no doubt will be received with excitement. In general, I find the study to be rigorous and carefully conducted. However, I am concerned that the limitations of the anatomical approach might have led to false negative conclusions and perhaps led to a circuit model that is premature. In addition, I am not convinced that the behavioral tests were appropriate for testing the involvement of the visual inputs in spatial navigation alone. As the authors themselves discuss, there is a mismatch between their results and what would be expected based on previous literature of a circuit directly impacting navigation. As they also suggest, fear and anxiety related behaviors, given the anatomical indicators could be more relevant. I think all sections need to be revised with these limitations in mind or additional experiments need to be conducted to more strongly support the arguments.

I have the following specific comments and suggestions:

What is the rationale for using PCP4 as a marker to delineate borders of dCE, vCE and ME? Could the authors annotate Supplementary fig 1e to show where the borders of these three divisions are on all of these sagittal sections? I am struggling to see a correspondence to commonly used references such as the Paxinos or the Allen Brain Atlas. This will most likely confuse all readers, so clarifying the rationale for the area boundary definitions plus showing the PCP4 based area definitions throughout the M-L extend of the EC would be ideal.

The conclusion in line 115 that a large proportion of L5 IN is involved in visual processing cannot be driven from the experiment conducted. The experiment shows that a large proportion of c-FOS+ neurons are interneurons. It does not assess what proportion of the total IN population these neurons make. I suggest revising this sentence.

Division of CA1 into a laterodorsal and mediodorsal part is unusual and confusing. I can see the challenge of labeling hippocampal divisions in the sagittal plane was an issue here. A quick look at where the so-called ldCA1 corresponds to in the coronal plane suggests that it is quite ventral, at a midpoint between dorsal and ventral CA1. It is harder to know where the so-called mdCA1 is as the image is cropped. I would suggest revising these naming conventions to make them more explicit by using coordinate positions and suggest including images of the whole slice for reference. I think it misguides the reader to think that there is cFOS expression in the dorsal CA1 when the ld-CA1 is referring to a middle position in the D-V axis.

One might think that the sudden light exposure could be a stress/fear inducing signal to the mouse. So the cFOS cells in the MEC could be representing a light induced fear instead of a pure response to light stimulation or has anything to do with spatial features of the MEC. This could explain why there is more expression in the vCA1 and the latero-dorsal CA1 (which appears to be close to dorsal/ventral CA1 boundary). In other words, what the cFOS signal represents here may not at all be purely visual signals so the conclusion of these regions collectively forming a “noncanonical visual pathway” (line 121,122) sounds a bit premature. Have the authors checked cFOS labeling in the amygdala or other fear associated structures? Can they exclude the possibility of anxiety in this experiment?

Supplementary fig 2g. Is this showing cellular labeling? I cannot distinguish cells in this image.

It is concluded that V1 is not sending signals to the EC. However I wonder if the injections in the V1 of the HSV were not representative of the entire V1. V1 is a very large structure, it cannot be covered by small deposits of virus as it was done in this study. And assessing the viral spread at the injection site is not possible because of the transsynaptic spreading properties of this virus. There is a possibility that parts of V1, that have not been included in this study, may project to the EC. Can the authors comment on how widely they sampled the V1? Can they quantify this in some way?

Similarly, the conclusions about the M-L and D-V coverage of trans synaptically or retrogradely labeled neurons needs a discussion about incomplete sampling due to the limitations of the tracing techniques. This will depend on what part/portion of the visual areas were injected. Throughout the study, the analysis of how much of the structures were covered by the viral injections are missing; this makes the conclusion about the topographical distribution of downstream neurons prone to being premature.

The design of the rabies retrograde labeling experiment is on the verge of making it inconclusive. A GFP reporter was used both to label anterograde transmission using the HSV as well as marking neurons that express TVA. I understand that this was done under the assumption that V1 does not project to the MEC therefore the authors do not expect transsynaptically labeled neurons expressing GFP. However, on the basis of my arguments above and since the assumptions were made on a negative result I think they cannot claim that where they saw TVA and rabies G overlap in the MEC (fig 3e) this is exclusive to the rabies viruses. I am also skeptical that the HSV viral strategy is capturing only anterograde transsynaptic neurons as it was previously described to also travel retrogradely. I don't think any of these technical limitations invalidate the conclusions but these need to be discussed and conclusions need to be toned accordingly.

Line 228 If I understand this correctly, the polysynaptic inhibition could be due to feedforward or feedback mechanisms, not just feedforward.

In the retrograde labeling experiment, I found the offset between the D-V position of the injection site in the superficial EC and the labeling of L5a neurons in the ventral EC quite odd (Figure 6b1). This is quite counterintuitive to all findings that show a D-V register within and between cortical areas in the brain. Perhaps the authors can support this result by including tracings of L5a neurons in the ventral EC/ME to show that their axons extend to the dorsal EC? They have the fills already; this would not require additional experiments. Also it is unclear to me why they did not inject their retro-dye in the more ventral superficial EC layers. Again my previous argument that the whole of V2M,V2L,V1 not just a small subsection need to be probed to draw more accurate conclusions about the topography of this circuit is strong.

What is the behavioral assay testing exactly? The timing of the inhibition suggests that what is being tested is memory retrieval performance. But the claim is that visual inputs are necessary for spatial navigation and online processing of visual cues. The way the test is designed it cannot disentangle if the reduction in task performance is due to interference with a memory retrieval circuit or whether the animals fail because of interrupting the visual information into the EC. The authors claim to suggest it is the latter but I think they need additional experiments if they want to show a pure effect on spatial navigation irrespective of memory.

In fact, the parts of the MEC that are shown to receive projection from the visual areas were previously reported to be poor in the number of cells with spatial features. Grid cells are primarily in the dorso-caudal EC, where visual inputs were not observed according to this study or polysynaptic circuits were not tested as this part of the EC is not in the same sagittal section as the ones used in the circuit mapping experiments. The possibility of this pathway being engaged in stress/anxiety type behaviors is strong, perhaps instead of pushing the conclusions to fit a spatial navigation narrative, it would be more appropriate to evaluate other options.

In some parts methods are very brief. What is “semi random” allocation of starting position in the MWM? What was the source of the viruses used? Source of CNO?

Reviewer #2 (Remarks to the Author):

In this manuscript, by using tracing experiments and electrophysiological recordings in brain slices, Shao et al. nicely showed a non-canonical visual pathway from V1-V2-MEC5a- hippocampal CA1/superficial MEC. By using fiber photometry and chemo/optogenetic experiments, they demonstrated that the V2-MEC5a pathway was important for spatial navigation in the water maze task. Overall, this is an interesting study. The experimental design is clear, the data is presented well, and the paper is well written. Before acceptance for publication, I have a few suggestions.

1, Please explain why dark rearing was required to see MEC5a neuronal responses (cfos expression or calcium responses by fiber photometry). Should the authors test a stronger light stimulation instead of dark rearing in both cfos and fiber photometry experiment? In the control group without dark rearing, did they significantly see cfos expression in the visual pathway, for example V1 or LGN?

2, The authors used fiber photometry to record the Ca²⁺ activities of V2L and V2M projecting MEC5a neurons with visual stimulation. It would be important to record the Ca²⁺ activities of these two pathways during the water maze training, to see if there is any activity change (or plasticity) over spatial task learning. This would further support the claim of importance of the pathway for spatial navigation and memory.

3, The authors used only one spatial navigation task (water maze task) to demonstrate the inhibition effect of V2-MECVa pathway on spatial memory recall. They should also repeat the experiment with other navigation tasks, such as radial maze or Barnes maze, to exclude the strong stimulation effect by water.

4, In Fig.1g, the author only provided the proportion of interneurons in cFos (+) neurons. They should also provide the proportion of excitatory neurons.

5, In general, the sample sizes in many experiments are too small (often $n = 3$ cells or mice, or 4 mice). These essential experiments, including electrophysiological recording, fiber photometry, and tracing, require verifications with more samples.

6, Provide the laser power of optogenetic inhibition in behaving mice.

Response to Reviewers

We appreciate the time and effort that the reviewers have dedicated to providing valuable feedback on our manuscript. We are grateful to the reviewers for their insightful comments. We have revised our manuscript and added new data and analyses to further improve the presentation and strengthen the conclusions of our work.

A point-by-point response to reviewers' concerns is provided below. The reviewers' comments are pasted verbatim below in italic font. Our responses are in blue font. Changes to the manuscript are highlighted in red font, which are also pasted below for the convenience of the reviewers. Additionally, line numbers and figure numbers cited below are those in the revised manuscript unless otherwise indicated.

Reviewer #1:

In this study, Shao et al., investigate how visual signals enter the Medial Entorhinal Cortex (MEC). Using a variety of tracing tools, they identify the direct and indirect targets of visual inputs to the MEC and draw very interesting and novel conclusions about the circuitry downstream of the visual inputs. Mainly, they find that projections to MEC primarily originate from the secondary visual area and directly target neurons in L5a which are responsive to light stimulation. Therefore, they conclude that the property of L5a neurons responding to light is due to the signals from V2. They further show that quite interesting V2M and V2L pathways target different subtypes of L5a neurons at different dendritic positions and move on to characterizing the downstream neurons and topographic relationships and synaptic properties of the connections between neurons downstream of the visual inputs. They then perform functional tests to probe the functional contribution of the visual inputs to MEC for spatial navigation.

The study is extremely timely and no doubt will be received with excitement. In general, I find the study to be rigorous and carefully conducted. However, I am concerned that the limitations of the anatomical approach might have led to false negative conclusions and perhaps led to a circuit model that is premature. In addition, I am not convinced that the behavioral tests were appropriate for testing the involvement of the visual inputs in spatial navigation alone. As the authors themselves discuss, there is a mismatch between their results and what would be expected based on previous literature of a circuit directly impacting navigation. As they also suggest, fear and anxiety related behaviors, given the anatomical indicators could be more relevant. I think all sections need to be revised with these limitations in mind or additional experiments need to be conducted to more strongly support the arguments.

We appreciate the reviewer's high evaluation of the present study and concerns regarding the limitations. We have conducted additional experiments to support our circuit model and confirm the behavioral results. We hope these additional results and the following response can address the reviewer's concerns.

I have the following specific comments and suggestions:

What is the rationale for using PCP4 as a marker to delineate borders of dCE, vCE and ME? Could the authors annotate Supplementary fig 1e to show where the borders of these three divisions are on all of these sagittal sections? I am struggling to see a correspondence to commonly used references such as the Paxinos or the Allen Brain Atlas. This will most likely confuse all readers, so clarifying the rationale for the area boundary definitions plus showing the PCP4 based area definitions throughout the M-L extend of the EC would be ideal.

1. We are sorry that the demarcation of MEC subdivisions was not presented clearly in the original manuscript. We have consulted resources including Doan et al., 2019, van Groen, 2001, Paxinos and Franklin's atlas and the Allen Brain Atlas (<http://mouse.brain-map.org/static/atlas>). The demarcation of MEC subdivisions was not completely consistent. In practice, we analyzed the EC region above the bottom edge of the hippocampal cross section and separated the dorsal 2/3 as CE. The CE-ME border was largely situated where PCP4 expression in L3 began to decline (Supplementary Fig. 1g). CE has not been further divided in previous studies and we did not aim to propose that CE should be further divided into subareas. However, we found that c-Fos⁺ cells evoked by light stimulation and L5a cells targeted by V2 were essentially not distributed in the dorsal-most part of CE (Fig. 1b and 1d; Fig. 2d-2e; Supplementary Fig. 3g-3n). Projecting fibers from V2 to CE also displayed a lower density in the dorsal-most part of CE (Fig. 2l). We therefore wanted to quantify these differences. In the revised manuscript, we counted cell numbers in the dorsal 1/4 (dCE) and the ventral 3/4 CE (vCE), respectively (Supplementary Fig. 1g). We are grateful for the reviewer's question, which made us reconsider our description in the manuscript. We have explicitly indicated that the dorsal 1/4 of CE was designated as dCE in the revised manuscript and have updated original Supplementary Fig. 1d with Supplementary Fig. 1g. As the reviewer suggested, we have also added borders in sagittal slices (Supplementary Fig. 1g).

Lines 109-112:

Interestingly, the distribution of c-Fos expression was layer and subregion dependent, with the **ventral CE** and ME expressing more c-Fos proteins, especially in layer 5 (L5; **Fig. 1d**). **We therefore counted cell numbers in the dorsal 1/4 (dCE) and the ventral 3/4 CE (vCE), respectively (Fig. 1b, Exp. 3; Supplementary Fig. 1g).**

Supplementary Fig. 1g:

g

Lines 1140-1141:

g, Sagittal sections showing that the MEC is divided into CE and ME subdivisions, with the former further divided into dorsal 1/4 (dCE) and ventral 3/4 parts (vCE).

The conclusion in line 115 that a large proportion of L5 IN is involved in visual processing cannot be driven from the experiment conducted. The experiment shows that a large proportion of c-FOS+ neurons are interneurons. It does not assess what proportion of the total IN population these neurons make. I suggest revising this sentence.

2. We agree with the reviewer's opinion. We have removed the expression from the sentence in the revised version.

Lines 115-117:

..., we found that most of c-Fos positive (c-Fos⁺) neurons in L5a were excitatory neurons and 52.0 ± 2.0% of c-Fos⁺ neurons in L5b were INs (**Fig. 1g-1h and Supplementary Fig. 1k-1l**).

Division of CA1 into a laterodorsal and mediodorsal part is unusual and confusing. I can see the challenge of labeling hippocampal divisions in the sagittal plane was an issue here. A quick look at where the so-called ldCA1 corresponds to in the coronal plane suggests that it is quite ventral, at a midpoint between dorsal and ventral CA1. It is harder to know where the so-called mdCA1 is as the image is cropped. I would suggest revising these naming conventions to make them more explicit by using coordinate positions and suggest including images of the whole slice for reference. I think it misguides the reader to think that there is cFOS expression in the dorsal CA1 when the ld-CA1 is referring to a middle position in the D-V axis.

3. We agree that ldCA1 in the original manuscript is at the intermediate part of CA1. Because this region is at the dorsal/top side in sagittal hippocampal slices and the hippocampus has been divided into dorsal 2/3 and ventral 1/3 parts in some study (Strange et al., 2014), we referred to it as lateral dorsal CA1, mimicking the term "lateral dorsal hippocampus" occasionally used by other papers (Kawakami et al., 1964; Tu et al., 2022). To address the reviewer's concern about confusion between ldCA1 and dorsal CA1, we have revised the names by changing ldCA1 to intermediate CA1 (iCA1) and mdCA1 to dorsal CA1 (dCA1) in the new version. The current nomenclature and demarcation were consistent with a previous study (Dong et al., 2009). We have also replaced original Supplementary Fig. 1g-1i with Supplementary Fig. 1o, displaying complete hippocampus and ML positions.

Lines 121, 218, 294, 298, 307, 546, 1041 and 1049; Fig. 1j, 3m, 5c and 6c₁; Supplementary Fig. 6d-6e:

... **iCA1** ...

Line 121; Fig. 1j and 3m:

... **dCA1** ...

Supplementary Figure 1o:

Lines 1153-1154:

o, Confocal images showing c-Fos expression in the visual cortical areas and the hippocampus for the mouse groups of Exp. 2 (dark rearing) and Exp. 3 (dark rearing + light stimulus).

One might think that the sudden light exposure could be a stress/fear inducing signal to the mouse. So the cFOS cells in the MEC could be representing a light induced fear instead of a pure response to light stimulation or has anything to do with spatial features of the MEC. This could explain why there is more expression in the vCA1 and the latero-dorsal CA1 (which appears to be close to dorsal/ventral CA1 boundary). In other words, what the cFOS signal represents here may not at all be purely visual signals so the conclusion of these regions collectively forming a “noncanonical visual pathway” (line 121,122) sounds a bit premature. Have the authors checked cFOS labeling in the amygdala or other fear associated structures? Can they exclude the possibility of anxiety in this experiment?

4. We agree that we could not exclude the possibility that circuits engaging in other functions like anxiety were also evoked in these light stimulation experiments. The significance of these initial experiments is that they indeed showed visual areas and the MEC were evoked (Fig. 1 and Supplementary Fig. 1) and therefore inspired the following study. Identification of the visual cortical-entorhinal pathway reported in the present study did not rely on these experiments but the following tracing experiments using a variety of anterograde and retrograde viral tracers as well as optogenetics. In addition, our data indeed suggested that c-Fos in the MEC were substantially due to direct responses to visual signals, because inhibition of V2 neurons significantly reduced c-Fos⁺ cells in the MEC (Fig. 7a-7c). Moreover, as the reviewer suggested, we have checked the LA, BLA and CeA regions of the amygdala, which are associated with fear and anxiety. However, we did not find significant enhancement of c-Fos expression (Supplementary Fig. 1m-1n).

Lines 122-124:

In addition, sudden exposure to light did not increase c-Fos expression in the lateral (LA), basolateral (BLA) and central amygdala (CeA), brain areas associated with fear and anxiety (**Supplementary Fig. 1m-1n**).

Supplementary Fig. 1m-1n:

Lines 1149-1152:

m, Coronal sections showing c-Fos expression in the lateral (LA), basolateral (BLA) and central amygdala (CeA) for the mouse groups of dark rearing or dark rearing with light stimuli.

n, Comparison of c-Fos⁺ cell numbers in the amygdala for the mouse groups of dark rearing ($n = 8$ slices from 4 mice) and dark rearing with light stimuli ($n = 8$ slices from 4 mice).

We agree that a claim of “non-canonical visual pathway” at the end of the first subsection of the Results section is premature. Indeed, we did not intend to make a conclusion but aimed to imply a hypothesis there. To address the reviewer’s concern, we have removed the expression from the sentence.

Line 122:

..., also responded to brief visual stimuli (**Fig. 1j** and **Supplementary Fig. 1o**).

Supplementary fig 2g. Is this showing cellular labeling? I cannot distinguish cells in this image.

5. We are sorry that the original figure did not show cells clearly. We have added enlarged images in the figure (Supplementary Fig. 2m in the revised manuscript).

Supplementary Fig. 2m:

It is concluded that V1 is not sending signals to the EC. However I wonder if the injections in the V1 of the HSV were not representative of the entire V1. V1 is a very large structure, it cannot be covered by small deposits of virus as it was done in this study. And assessing the viral spread at the injection site is not possible because of the transsynaptic spreading properties of this virus. There is a possibility that parts of V1, that have not been included in this study, may project to the EC. Can the authors comment on how widely they sampled the V1? Can they quantify this in some way?

6. We agree that in our original experiments, the extent of injection site in V1 was limited. To address the reviewer's concern, we injected HSV at multiple sites to cover the most extent of V1. We did not find EGFP⁺ cells at 48 HPI in MEC (Supplementary Fig. 2e-2g). Consistently, injecting AAV2/9-CAG-mCherry at multiple sites of V1, we found no projecting fibers of V1 in MEC (Supplementary Fig. 2h-2j). Moreover, in retrograde tracing experiments with RV, labeled presynaptic neurons were essentially in V2 (Supplementary Fig. 5a-5d). All these data indicated that V2 was the major direct source of visual inputs to the MEC. Nonetheless, we do not aim to deny that rare V1 inputs to the MEC might exist anyway. Thus, we wrote "These results together imply that direct synaptic connections from V1 to MEC are rare if not absent, ..." in lines 142-143 (lines 137-138 in the original version).

Lines 139-142:

Given the large size of V1, we also injected H129-hUbC-HBEGFP at multiple sites of V1 and got consistent results (Supplementary Fig. 2e-2g). These data were further supported by observation that the V1 hardly project to the MEC (Supplementary Fig. 2h-2j).

Supplementary Fig. 2e-2j:

Lines 1167-1173:

e, Schematic illustrating injection of HSV1 H129 at multiple sites of V1.

f, Sagittal sections showing the HSV1 H129 expression at 48 HPI or 60 HPI within the MEC.

g, Comparison of HSV1 H129 expression within the MEC L5a at 48 HPI and 60 HPI ($n = 8$ slices from 4 mice/group). *** $P < 0.001$, Mann-Whitney U test.

h, Schematic illustrating injection of AAV2/9-CAG-mCherry at multiple sites of V1.

i, Sagittal sections showing the AAV2/9-CAG-mCherry expression in V1 at different ML positions.

j, Sagittal sections showing that V1 projections were not present in the MEC.

Supplementary Fig. 5c:

Line 1223:

c, Sagittal sections showing the RV expression in V1 at different ML positions.

Similarly, the conclusions about the M-L and D-V coverage of trans synaptically or retrogradely labeled neurons needs a discussion about incomplete sampling due to the limitations of the tracing techniques. This will depend on what part/portion of the visual areas were injected. Throughout the study, the analysis of how much of the structures were covered by the viral injections are missing; this makes the conclusion about the topographical distribution of downstream neurons prone to being premature.

7. We agree that V2 is a large area and the extent of injection site in V2 was limited, therefore only subpopulation of L5a neurons in the MEC was labeled in the experiments. However, we believe that their distribution essentially reflects the ML and DV coverage of L5a neurons targeted by the entire V2. First, in the present study, we at least injected virus at two sites of V2, i.e. sites in V2M and V2L, respectively. They were far from each other and separated by V1. Even so, they both mainly targeted vCE L5a neurons and these neurons had a similar ML distribution. Moreover, these transsynaptically labeled neurons largely have a consistent distribution along the ML or DV axis with c-Fos⁺ neurons. Second, V1 targets to all subareas of V2 (Wang and Burkhalter, 2007). Thus, in experiments where

HSV was injected into V1 and samples were collected at 60 HPI (Fig. 2a-2c; Supplementary Fig. 2e-2g; Fig 6a-6d), labeled neurons in the MEC should contain neurons targeted by multiple subareas of V2. These labeled neurons were also mainly distributed in vCE L5a. Third, retrograde RV labeling data not only corroborated anterograde results, but also got insight of distribution of presynaptic neurons in the whole V2, showing multiple subareas of V2 targeted vCE L5a neurons. Taking these data together, we therefore believe that it is unlikely that there are substantial V2-targeted L5a neurons distributed in a distinct MEC subregion missed in our tracing experiments. Besides, we also conducted additional experiments with HSV injected at a more medial and anterior position in V2L and V2M and confirmed that vCE L5a neurons were the main target of V2 (Supplementary Fig. 3k-3n).

Lines 156-158:

Anterograde trans-monosynaptic tracing by H129-hUbC-HBEGFP, injected either at similar positions or more medially and anteriorly in V2L and V2M, corroborated vCE L5a as the main target (Supplementary Fig. 3g-3n).

Supplementary Fig. 3k-3n:

Lines 1203-1206:

k, HSV1 H129 was injected more medially and anteriorly in V2L compared to **g**.

l, Comparison of HSV expression within dCE, vCE, ME of MEC L5a at 48 HPI ($n = 6$ slices from 3 mice).

m, HSV1 H129 was injected more medially and anteriorly in V2M compared to **i**.

n, Comparison of HSV expression within dCE, vCE, ME of MEC L5a at 48 HPI ($n = 6$ slices from 3 mice).

The design of the rabies retrograde labeling experiment is on the verge of making it inconclusive. A GFP reporter was used both to label anterograde transmission using the HSV as well as marking neurons that express TVA. I understand that this was done under the assumption that V1 does not project to the MEC therefore the authors do not expect transsynaptically labeled neurons expressing GFP. However, on the basis of my arguments above and since the assumptions were made on a negative result I think they cannot claim that where they saw TVA and rabies G overlap in the MEC (fig 3e) this is exclusive to the rabies viruses. I am also skeptical that the HSV viral strategy is capturing only anterograde transsynaptic neurons as it was previously described to also travel retrogradely. I don't think any of these technical limitations invalidate the conclusions but these need to be discussed and conclusions need to be toned accordingly.

8. We acknowledge technical limitations of viral tools. Therefore, we designed independent experiments using combinations of various viruses so that multiple experiments together led a solid conclusion. In the manuscript, we presented the data combining RV and HSV after we had showed that injecting HSV alone in V1 did not label postsynaptic neurons in MEC at 48 HPI (Fig. 2b). Although the injection site only covered a limited area of V1, it is unlikely that some postsynaptic neurons missed by experiments of Fig. 2a-2c were labeled in experiments of Fig. 3d-3f since in both groups of experiments HSV was basically injected in the same position of V1. Furthermore, as mentioned above, our new data in Supplementary Fig. 2e-2j should provide further support. On the other hand, EGFP cells in the MEC in Fig. 3e could not be due to retrograde labeling by HSV in V1. For HSV, the direction of transsynaptic transport is strain-dependent. H129 strain moves only anterogradely (Zemanick et al., 1991; Sun et al., 1996; Beier et al., 2011). In addition, we have also presented results without HSV injected in V1 in Supplementary Fig. 5, validating the RV tracing. We noticed that the reviewer also thought the conclusions were validate but suggested we revise our expression. In the revised manuscript, we have adjusted the positions of texts referring to Supplementary Fig. 5 and Fig. 3d-3f and modified sentences accordingly.

Line 206-213:

Finally, on day 24 brain tissues were collected (**Supplementary Fig. 5a**). In sagittal sections, double-labeled neurons (starter neurons) were found in MEC L5a as expected (**Supplementary Fig. 5b**). Consistent with anterograde tracing data, presynaptic neurons in V2L and V2M, rather than V1, were observed (**Supplementary Fig. 5b-5d**). After validating the RV tracing, we also injected anterograde tracer H129-hUbC-HBEGFP into V1 two days before tissue collection (**Fig. 3d**). In tangential sections through flat-mounted cortex, double-labeled neurons present in MEC L5a (**Fig. 3e**) were putative starter neurons since injection of HSV1 H129 in V1 did not label MEC cells at 48 HPI (**Fig. 2b-2c; Supplementary Fig. 2e-2g**).

Line 228 If I understand this correctly, the polysynaptic inhibition could be due to feedforward or feedback mechanisms, not just feedforward.

9. Disynaptic feedforward inhibition was more likely, but feedback inhibition involving more synapses was also possible. We have revised the sentence.

Line 235:

... , indicating that the IPSCs were due to **polysynaptic** inhibition mediated by local inhibitory neurons.

In the retrograde labeling experiment, I found the offset between the D-V position of the injection site in the superficial EC and the labeling of L5a neurons in the ventral EC quite odd (Figure 6b1). This is quite counterintuitive to all findings that show a D-V register within and between cortical areas in the brain. Perhaps the authors can support this result by including tracings of L5a neurons in the ventral EC/ME to show that their

axons extend to the dorsal EC? They have the fills already; this would not require additional experiments. Also it is unclear to me why they did not inject their retro-dye in the more ventral superficial EC layers. Again my previous argument that the whole of V2M, V2L, V1 not just a small subsection need to be probed to draw more accurate conclusions about the topography of this circuit is strong.

10. Thanks for the reviewer's suggestion. We have labeled ME L5a neurons by EGFP and demonstrated that their axons extended to superficial layers of the dorsal MEC (Supplementary Fig. 8a-8b). We injected AAV2retro in the dorsal superficial MEC because there are more sharply tuned grid cells and we wanted to examine direct projections from CE L5a neurons. For the concern about the injection region of V2 and V1, please see our response to Points 6 and 7.

Supplementary Fig. 8a-8b:

Lines 1275-1277:

a, Schematic illustrating viral injection strategy to label ME L5a neurons.

b, Confocal images showing the labeled ME L5a neurons and their axons in the dorsal MEC. Partial axons in the dorsal MEC were indicated by arrowheads. Scale bar, 200 μ m (left), 100 μ m (right).

What is the behavioral assay testing exactly? The timing of the inhibition suggests that what is being tested is memory retrieval performance. But the claim is that visual inputs are necessary for spatial navigation and online processing of visual cues. The way the test is designed it cannot disentangle if the reduction in task performance is due to interference with a memory retrieval circuit or whether the animals fail because of interrupting the visual information into the EC. The authors claim to suggest it is the latter but I think they need additional experiments if they want to show a pure effect on spatial navigation irrespective of memory.

In fact, the parts of the MEC that are shown to receive projection from the visual areas were previously reported to be poor in the number of cells with spatial features. Grid cells are primarily in the dorso-caudal EC, where visual inputs were not observed according to this study or polysynaptic circuits were not tested as this part of the EC is not in the same sagittal section as the ones used in the circuit mapping experiments. The possibility of this pathway being engaged in stress/anxiety type behaviors is strong, perhaps instead of pushing the conclusions to fit a spatial navigation narrative, it would be more appropriate to evaluate other options.

11. Morris water maze (MWM) is a memory-guided spatial navigation task. Spatial memory retrieval is a critical component of the navigation task and requires sustained visual inputs. Navigation using the

spatial memory strategy relies on the hippocampal-entorhinal system (Dahmani and Bohbot, 2020). In this context, the navigation circuit and the memory circuit are the same. Navigation and memory have been proposed to employ the same neuronal algorithms (Buzsáki and Moser, 2013) and observed to elicit similar electrical activity in human brain (Solomon et al., 2019). In the present study, we first inhibited MEC L5a neurons receiving V2 projections and found impaired performance of animals in the MWM test (Fig. 8a-8d). However, MEC L5a neurons gate bidirectional cortical-hippocampal communications. They also relay hippocampal outputs to the neocortex. To further isolate the effect of V2→MEC L5a pathway, we next directly inhibited the projecting fibers from V2M/V2L in the MWM test (Fig. 8e-8h). The results showed that interrupting the visual inputs into the EC impaired the task performance. In the revised manuscript, we have conducted additional experiments with the Barnes maze (Fig. 8i-8k), which further corroborated the findings in MWM test. In addition, we have also used fiber photometry to monitor the activity of MEC L5a neurons targeted by V2M/V2L during the MWM task and found enhancement in neural activity over spatial task learning (Fig 8l-8p and Supplementary Fig. 11a-11d). Moreover, We have also found that the L5a Ca²⁺ activity of well-trained mice in spatial tasks required sustained visual inputs (Fig 8l-8p and Supplementary Fig. 11a-11d).

Grid cells were first discovered in the dorso-caudal EC (Hafting et al., 2005), but they are present throughout the DV axis of the MEC with their firing fields having a DV gradient (Stensola et al., 2012). Grid cells were also found in all layers of the MEC, but L2 contains the largest density of pure grid cells (Sargolini et al., 2006). Therefore, we have tested the polysynaptic circuit V2→MEC L5a→MEC L2 by tracing and optogenetic experiments (Fig. 5l and 6b). In addition, we also demonstrated connections from V2-targeted MEC L5a to CA1 (Fig. 5k and 6c-6d), which is also pivotal for spatial cognition.

In summary, our data have shown that this pathway routes visual input to brain areas critical for spatial cognition and is necessary in memory-guided spatial navigation. Thus, we think the conclusion is reasonable. Nonetheless, we agree with the reviewer that this pathway could have other functions, such as engaging in stress/anxiety type behaviors. Indeed, we mentioned this possibility in the last paragraph of the Discussion section and proposed it as a future research direction.

Lines 87-91:

Finally, we employed chemogenetic inhibition of MEC L5a neurons receiving V2 projections **in the Morris water maze (MWM) test** or optogenetic inhibition of projecting axons from V2 **in MWM and the Barnes maze (BM) tests**. Both manipulations impaired the performance of animals in navigation **tasks and the associated activity of MEC L5a neurons required sustained visual inputs**.

Lines 372-377:

During training sessions in MWM **and BM (Fig. 8e and 8i)**, there was no difference in learning ability between the sham control and eNpHR3.0 groups of mice (**Supplementary Fig. 10f and 10g**). However, when applying the yellow laser to well-trained mice, optogenetic suppression of either V2L or V2M afferent fibers in the MEC disrupted their ability to directly navigate to the target (**Fig. 8g-8h and 8j-8k**).

In the sham control group, mice that expressed EGFP in V2L or V2M showed normal escape latencies during the laser was on (**Fig. 8g-8h and 8j-8k**).

Lines 471-473:

Our results of **MWM and BM** demonstrated that both V2M and V2L inputs were indispensable for spatial navigation and the associated neural activity of MEC L5a.

Lines 682-693:

Barnes maze

Mice were also tested for spatial cognition in the Barnes maze (90 cm diameter, 20 holes, 5 cm hole diameter, 60 cm height). Four visual cues in different shapes and colors were placed around the maze. All sessions were recorded using an infrared camera. A bright light was affixed above the maze to motivate the escape behavior. The Barnes maze was cleaned before and after each testing trial with 75% ethanol. During the training phase (Day 1-5), mice were tested for 2 trials per day. For each trial, mice were first placed in an opaque cup in the center of the maze for 5 s. Then, the cup was lifted, and mice were allowed to explore the maze freely for 3 min. After finding the escape box, mice were allowed to rest in the box for 2 min before returning to cages. The animals that failed to find the escape box in 3 min were slowly guided to the target hole and gently pushed into the escape box if they did not enter by themselves. The probe trial was performed on day 6. For each trial, mice were allowed to explore the maze for 1 min. Primary latency was defined as the time to first visit the target hole and was recorded over the trainings.

Lines 697-698:

The recovered mice received a 6-day spatial acquisition training in Morris water maze or a 5-day spatial acquisition training in Barnes maze as described above.

Fig. 8i-8k:

Lines 1104-1111:

i, Schematic illustrating optogenetic manipulation via bilaterally implanted optical fibers during BM tests. Cyan circle denotes the escape hole.

j, Representative movement trajectory for well-trained mice of control and eNpHR3.0 groups (left: viral injection in V2M; right: viral injection in V2L) during consecutive light-off, light-on, and light-off trials.

k, Left: summary plot exhibiting the effect of light-induced eNpHR3.0 activation on primary latency in the V2M-eNpHR3.0 group ($n = 7$ mice) and the V2M-EGFP group ($n = 7$ mice). Right: summary plot exhibiting the effect of light-induced eNpHR3.0 activation on primary latency in the V2L-eNpHR3.0 group ($n = 7$ mice) or the V2L-EGFP group ($n = 7$ mice).

Supplementary Fig. 10g:

Lines 1325-1326:

g, Primary latencies of the control group and the eNpHR3.0 group in the Barnes maze test over five consecutive training days.

Lines 380-398:

To investigate the activity of vCE L5a neurons receiving V2M or V2L projections during spatial navigation, we then used fiber photometry to record Ca^{2+} signals in freely moving mice during the MWM task (**Fig. 8l**). After several days of training, the mice were proficient in finding the hidden platform quickly on day 4 (**Fig. 8n** and **8p**). Interestingly, the activity of vCE L5a neurons receiving V2M or V2L projections showed enhancement associated with navigation experience. During the initial exploratory navigation on day 1, an increase in Ca^{2+} activity was barely detectable (**Fig. 8m-8p**). In contrast, on day 4 vCE L5a neurons receiving V2M or V2L projections produced a large and sustained Ca^{2+} signal (**Fig. 8m-8p**). Notably, on day 4 the Ca^{2+} signals rose at the beginning of the navigation and decayed when the hidden platform was reached (**Supplementary Fig. 11a-11b**), indicating that vCE L5a neurons receiving V2M or V2L projections were involved in the navigation task. Next, to determine whether the activity of vCE L5a neurons involved in the navigation task requires visual inputs, on day 5 we gave mice 1-s lighting during the initial phase of navigation and then turned off the ambient light or turned off the ambient light throughout the navigation. In the dark condition, the mice had difficulty in navigating to the platform and the activity of vCE L5a neurons receiving V2M or V2L projections was significantly decreased (**Fig. 8m-8p; Supplementary Fig. 11a-11d**). In the 1-s lighting condition, access to visual cues during the initial phase only mildly improved the performance of mice and the activity of the vCE L5a neurons receiving V2M or V2L projections declined after the light was turned off despite its normal initial rising phase (**Fig.**

8m-8p; Supplementary Fig. 11a-11d). These results indicate that visual inputs are necessary for the activity of vCE L5a neurons involved in the navigation during the MWM task. To summarize, the above behavioral data together reveal a critical role of the V2→MEC L5a pathway in memory-guided spatial navigation.

Lines 659-662:

To verify L5a neurons were involved in the MWM task, mice were fitted with an optical fiber to swim in the pool. During each trial, mice were allowed to search the platform for 60 s. An infrared camera was used so that recordings could be made even in darkness. All the fluorescence data were segmented and aligned to the onset of light stimuli or MWM task within individual trials.

Fig. 8l-8p:

Lines 1112-1116:

l, Schematic illustrating fiber photometry recording of Ca^{2+} signals during MWM tests.

m and **o**, Examples of Ca^{2+} signals of vCE L5a neurons receiving V2M projections (**m**) or V2L projections (**o**) on training days 1- 4 and in the 1-s lighting or the dark condition on day 5.

n and **p**, Plot of the escape latencies (black) and their corresponding amplitudes of Ca^{2+} signals (green) at different training days ($n = 5$ mice for each group).

Supplementary Fig. 11a-11d:

Lines 1331-1336:

Supplementary Figure 11, related to Figure 8. Neural activity of MEC L5a neurons in the MWM tests.

a-b, Examples of the swim path (top) and the corresponding Ca^{2+} signals (bottom) in the vCE L5a neurons receiving V2M (a) or V2L (b) projections on day 1, day 4, and day 5 with 1-s light or in dark. The Ca^{2+} signal traces and their corresponding swim paths are marked in red.

c-d, Mean and SEM of Ca^{2+} signals in the vCE L5a neurons receiving V2M (c, $n = 5$ mice) or V2L (d, $n = 5$ mice) projections.

In some parts methods are very brief. What is “semi random” allocation of starting position in the MWM? What was the source of the viruses used? Source of CNO?

12. We are sorry that “semi-random” was incorrectly used in the original manuscript. In three trials, we released the animals in three different quadrants other than the target quadrant, respectively. We are grateful for the reviewer’s careful reading. We have revised our expression. In addition, we have added the required information for viruses and CNO in the revised manuscript.

Lines 575-576:

All viruses and dye were purchased from BrainVTA (Wuhan, China), Brain Case Biotech (Shenzhen, China) and OBiO Technology (Shanghai, China).

Line 628:

CNO (5 μM ; APEX BIO, A3317)

Lines 657 and 680:

... CNO (1 mg/kg; APEX BIO, A3317) ...

Line 672:

... containing 3 trials with different start positions in three quadrants other than the target quadrant, respectively)

Reviewer #2:

In this manuscript, by using tracing experiments and electrophysiological recordings in brain slices, Shao et al. nicely showed a non-canonical visual pathway from V1-V2-MEC5a- hippocampal CA1/superficial MEC. By using fiber photometry and chemo/optogenetic experiments, they demonstrated that the V2-MEC5a pathway was important for spatial navigation in the water maze task. Overall, this is an interesting study. The experimental design is clear, the data is presented well, and the paper is well written. Before acceptance for publication, I have a few suggestions.

We appreciate the reviewer's high evaluation of the present study and the thoughtful suggestions. We have conducted additional experiments as the reviewer suggested. We hope these additional results and the following response can address the reviewer's concerns.

1, Please explain why dark rearing was required to see MEC5a neuronal responses (cfos expression or calcium responses by fiber photometry). Should the authors test a stronger light stimulation instead of dark rearing in both cfos and fiber photometry experiment? In the control group without dark rearing, did they significantly see cfos expression in the visual pathway, for example V1 or LGN?

At the beginning of our study, we also expected to see increased c-Fos expression in normal rearing mice subject to a light stimulus. However, the results were negative (Fig. 1a-c). We could only detect c-Fos expression when mice were reared in dark for enough time (Supplementary Fig. 1d). In the visual pathway, the results were the same (Supplementary Fig. 1h-1i). Similar results have also been reported in other studies (Chaudhuri et al., 2000; Sato et al., 2000). The underlying mechanism is unclear, but it was suggested that daily repeated exposure to ambient light may well suppress the c-Fos induction (Sato et al., 2000). As to calcium responses by fiber photometry, we did not rear animal in dark but only recorded in dark to avoid possible interference from ambient light.

As the reviewer suggested, we have tried a stronger light stimulation (5200 lx vs. 550 lx used before) but found that c-Fos expression was still on a low level (Supplementary Fig. 1a-1c), suggesting that the negative result in Exp. 1 of Fig. 1a-1c was not due to an insufficient light intensity.

Lines 99-100:

Even with a much stronger light stimulus (5200 lx, 2 min), c-Fos expression remained low (**Supplementary Fig. 1a-1c**).

Supplementary Fig. 1a-1c:

Lines 1130-1133:

- a**, Schematic illustrating that normally reared mice were given a stronger light stimulus (5200 lx, 2 min).
- b**, Confocal image showing the corresponding c-Fos expression.
- c**, Comparison of c-Fos⁺ cell numbers in the MEC for the mouse group of dark rearing with light stimuli (*n* = 7 mice) and the mouse group of normal rearing with stronger light stimuli (*n* = 7 mice).

Lines 105-107:

Compared with those receiving no light stimuli, mice subjected to a light stimulus had a significantly higher c-Fos expression in the MEC **as well as visual areas** (Fig. 1a-1b, Exp. 2 vs. Exp. 3; Fig. 1c and Supplementary Fig. 1h-1i).

Supplementary Fig. 1h-1i:

Lines 1142-1145:

- h**, Confocal image showing c-Fos expression in the visual cortex for the mouse group of normal rearing with light stimuli.
- i**, Comparison of c-Fos⁺ cell numbers in the MEC for the mouse groups of dark rearing with light stimuli (*n* = 6 mice) and normal rearing with light stimuli (*n* = 6 mice).

2, The authors used fiber photometry to record the Ca²⁺ activities of V2L and V2M projecting MEC5a neurons with visual stimulation. It would be important to record the Ca²⁺ activities of these two pathways during the water maze training, to see if there is any activity change (or plasticity) over spatial task learning. This would further support the claim of importance of the pathway for spatial navigation and memory.

Thanks for the reviewer's suggestion. We have conducted the suggested experiments and demonstrated an enhancement in Ca^{2+} activity of these two pathways over spatial task learning (Fig. 8l-8p and Supplementary Fig. 11a-11d). Moreover, We also found that the L5a Ca^{2+} activity of well-trained mice in spatial tasks required sustained visual inputs (Fig. 8l-8p and Supplementary Fig. 11a-11d).

Lines 380-398:

To investigate the activity of vCE L5a neurons receiving V2M or V2L projections during spatial navigation, we then used fiber photometry to record Ca^{2+} signals in freely moving mice during the MWM task (Fig. 8l). After several days of training, the mice were proficient in finding the hidden platform quickly on day 4 (Fig. 8n and 8p). Interestingly, the activity of vCE L5a neurons receiving V2M or V2L projections showed enhancement associated with navigation experience. During the initial exploratory navigation on day 1, an increase in Ca^{2+} activity was barely detectable (Fig. 8m-8p). In contrast, on day 4 vCE L5a neurons receiving V2M or V2L projections produced a large and sustained Ca^{2+} signal (Fig. 8m-8p). Notably, on day 4 the Ca^{2+} signals rose at the beginning of the navigation and decayed when the hidden platform was reached (Supplementary Fig. 11a-11b), indicating that vCE L5a neurons receiving V2M or V2L projections were involved in the navigation task. Next, to determine whether the activity of vCE L5a neurons involved in the navigation task requires visual inputs, on day 5 we gave mice 1-s lighting during the initial phase of navigation and then turned off the ambient light or turned off the ambient light throughout the navigation. In the dark condition, the mice had difficulty in navigating to the platform and the activity of vCE L5a neurons receiving V2M or V2L projections was significantly decreased (Fig. 8m-8p; Supplementary Fig. 11a-11d). In the 1-s lighting condition, access to visual cues during the initial phase only mildly improved the performance of mice and the activity of the vCE L5a neurons receiving V2M or V2L projections declined after the light was turned off despite its normal initial rising phase (Fig. 8m-8p; Supplementary Fig. 11a-11d). These results indicate that visual inputs are necessary for the activity of vCE L5a neurons involved in the navigation during the MWM task. To summarize, the above behavioral data together reveal a critical role of the V2→MEC L5a pathway in memory-guided spatial navigation.

Lines 659-662:

To verify L5a neurons were involved in the MWM task, mice were fitted with an optical fiber to swim in the pool. During each trial, mice were allowed to search the platform for 60 s. An infrared camera was used so that recordings could be made even in darkness. All the fluorescence data were segmented and aligned to the onset of light stimuli or MWM task within individual trials.

Fig. 8l-8p:

Lines 1112-1116:

I, Schematic illustrating fiber photometry recording of Ca^{2+} signals during MWM tests.

m and **o**, Examples of Ca^{2+} signals of vCE L5a neurons receiving V2M projections (**m**) or V2L projections (**o**) on training days 1- 4 and in the 1-s lighting or the dark condition on day 5.

n and **p**, Plot of the escape latencies (black) and their corresponding amplitudes of Ca^{2+} signals (green) at different training days ($n = 5$ mice for each group).

Supplementary Fig. 11a-11d:

Lines 1331-1336:

Supplementary Figure 11, related to Figure 8. Neural activity of MEC L5a neurons in the MWM tests.

a-b, Examples of the swim path (top) and the corresponding Ca^{2+} signals (bottom) in the vCE L5a neurons receiving V2M (**a**) or V2L (**b**) projections on day 1, day 4, and day 5 with 1-s light or in dark. The Ca^{2+} signal traces and their corresponding swim paths are marked in red.

c-d, Mean and SEM of Ca^{2+} signals in the vCE L5a neurons receiving V2M (**c**, $n = 5$ mice) or V2L (**d**, $n = 5$ mice) projections.

3, The authors used only one spatial navigation task (water maze task) to demonstrate the inhibition effect of V2-MECVa pathway on spatial memory recall. They should also repeat the experiment with other navigation tasks,

such as radial maze or Barnes maze, to exclude the strong stimulation effect by water.

Thanks for the reviewer's suggestion. We have conducted additional experiments with Barnes maze (Fig. 8i-8k). The results corroborated our findings in water maze task.

Lines 87-91:

Finally, we employed chemogenetic inhibition of MEC L5a neurons receiving V2 projections **in the Morris water maze (MWM) test** or optogenetic inhibition of projecting axons from V2 **in MWM and the Barnes maze (BM) tests**. Both manipulations impaired the performance of animals in navigation **tasks and the associated activity of MEC L5a neurons required sustained visual inputs**.

Lines 372-377:

During training sessions in MWM **and BM (Fig. 8e and 8i)**, there was no difference in learning ability between the sham control and eNpHR3.0 groups of mice (**Supplementary Fig. 10f and 10g**). However, when applying the yellow laser to well-trained mice, optogenetic suppression of either V2L or V2M afferent fibers in the MEC disrupted their ability to directly navigate to the target (**Fig. 8g-8h and 8j-8k**). In the sham control group, mice that expressed EGFP in V2L or V2M showed normal escape latencies during the laser was on (**Fig. 8g-8h and 8j-8k**).

Lines 471-473:

Our results **of MWM and BM** demonstrated that both V2M and V2L inputs were indispensable for spatial navigation **and the associated neural activity of MEC L5a**.

Lines 682-693:

Barnes maze

Mice were also tested for spatial cognition in the Barnes maze (90 cm diameter, 20 holes, 5 cm hole diameter, 60 cm height). Four visual cues in different shapes and colors were placed around the maze. All sessions were recorded using an infrared camera. A bright light was affixed above the maze to motivate the escape behavior. The Barnes maze was cleaned before and after each testing trial with 75% ethanol. During the training phase (Day 1-5), mice were tested for 2 trials per day. For each trial, mice were first placed in an opaque cup in the center of the maze for 5 s. Then, the cup was lifted, and mice were allowed to explore the maze freely for 3 min. After finding the escape box, mice were allowed to rest in the box for 2 min before returning to cages. The animals that failed to find the escape box in 3 min were slowly guided to the target hole and gently pushed into the escape box if they did not enter by themselves. The probe trial was performed on day 6. For each trial, mice were allowed to explore the maze for 1 min. Primary latency was defined as the time to first visit the target hole and was recorded over the trainings.

Lines 697-698:

The recovered mice received a 6-day spatial acquisition training in Morris water maze **or a 5-day spatial acquisition training in Barnes maze** as described above.

Fig. 8i-8k:

Lines 1104-1111:

i, Schematic illustrating optogenetic manipulation via bilaterally implanted optical fibers during BM tests. Cyan circle denotes the escape hole.

j, Representative movement trajectory for well-trained mice of control and eNpHR3.0 groups (left: viral injection in V2M; right: viral injection in V2L) during consecutive light-off, light-on, and light-off trials.

k, Left: summary plot exhibiting the effect of light-induced eNpHR3.0 activation on primary latency in the V2M-eNpHR3.0 group ($n = 7$ mice) and the V2M-EGFP group ($n = 7$ mice). Right: summary plot exhibiting the effect of light-induced eNpHR3.0 activation on primary latency in the V2L-eNpHR3.0 group ($n = 7$ mice) or the V2L-EGFP group ($n = 7$ mice).

Supplementary Fig. 10g:

Lines 1325-1326:

g, Primary latencies of the control group and the eNpHR3.0 group in the Barnes maze test over five consecutive training days.

4, In Fig. 1g, the author only provided the proportion of interneurons in cFos (+) neurons. They should also provide the proportion of excitatory neurons.

The proportion of excitatory neurons in c-Fos⁺ neurons is expected to be complementary to the proportion of interneurons in c-Fos⁺ neurons. To directly address the reviewer's concern, we have used CaMKII antibody to verify the proportion of excitatory neurons in L5a c-Fos⁺ neurons (Supplementary Fig. 1k-1l).

Lines 114-117:

Furthermore, using CaMKII antibody or Vgat^{Cre} x Ai9 transgenic mice where GABAergic interneurons (INs) are fluorescently labeled, we found that most of c-Fos positive (c-Fos⁺) neurons in L5a were excitatory neurons and $52.0 \pm 2.0\%$ of c-Fos⁺ neurons in L5b were INs (Fig. 1g-1h and Supplementary Fig. 1k-1l).

Supplementary Fig. 1k-1l:

Lines 1147-1148:

k, Confocal images showing c-Fos⁺ cells co-labeled with CaMKII antibody in MEC L5.

l, Proportion of co-labeled cells in c-Fos⁺ cells for MEC L5a and L5b ($n = 8$ slices from 4 mice).

5, In general, the sample sizes in many experiments are too small (often $n = 3$ cells or mice, or 4 mice). These essential experiments, including electrophysiological recording, fiber photometry, and tracing, require verifications with more samples.

Thanks for the reviewer's suggestion. We have added more samples in experiments of Fig. 1i-1j, 2j, 4l, 5c, 5e, 5h-5j, 5l, 6f and 7f-7g and Supplementary Fig. 1d, 1f and 7f-7h. The sample sizes in this study are comparable to other studies (Sürmeli et al., 2015; Huang et al., 2017; Ohara et al., 2018; Qin et al., 2018; Shang et al., 2018; Huang et al., 2019; Kecskés et al., 2020; Tsoi et al., 2022; Ohara et al., 2023).

6, Provide the laser power of optogenetic inhibition in behaving mice.

We have added the information in the revised manuscript.

Line 701:

Light power used in the optogenetic inhibition was 18 mW.

REFERENCES

- Beier KT, Saunders A, Oldenburg IA, Miyamichi K, Akhtar N, Luo L, Whelan SP, Sabatini B, Cepko CL (2011) Anterograde or retrograde transsynaptic labeling of CNS neurons with vesicular stomatitis virus vectors. *Proceedings of the National Academy of Sciences of the United States of America* 108:15414-15419.
- Buzsáki G, Moser EI (2013) Memory, navigation and theta rhythm in the hippocampal-entorhinal system. *Nature Neuroscience* 16:130-138.
- Chaudhuri A, Zangenehpour S, Rahbar-Dehgan F, Ye F (2000) Molecular maps of neural activity and quiescence. *Acta Neurobiologiae Experimentalis* 60:403-410.
- Dahmani L, Bohbot VD (2020) Habitual use of GPS negatively impacts spatial memory during self-guided navigation. *Scientific Reports* 10:6310.
- Doan TP, Lagartos-Donate MJ, Nilssen ES, Ohara S, Witter MP (2019) Convergent projections from perirhinal and postrhinal cortices suggest a multisensory nature of lateral, but not medial, entorhinal cortex. *Cell Reports* 29:617-627.
- Dong HW, Swanson LW, Chen L, Fanselow MS, Toga AW (2009) Genomic-anatomic evidence for distinct functional domains in hippocampal field CA1. *Proceedings of the National Academy of Sciences of the United States of America* 106:11794-11799.
- Hafting T, Fyhn M, Molden S, Moser MB, Moser EI (2005) Microstructure of a spatial map in the entorhinal cortex. *Nature* 436:801-806.
- Huang L, Yuan T, Tan M, Xi Y, Hu Y, Tao Q, Zhao Z, Zheng J, Han Y, Xu F, Luo M, Sollars PJ, Pu M, Pickard GE, So KF, Ren C (2017) A retinorecipient projection regulates serotonergic activity and looming-evoked defensive behaviour. *Nature Communications* 8:14908.
- Huang L, Xi Y, Peng Y, Yang Y, Huang X, Fu Y, Tao Q, Xiao J, Yuan T, An K, Zhao H, Pu M, Xu F, Xue T, Luo M, So KF, Ren C (2019) A visual circuit related to habenula underlies the antidepressive effects of light therapy. *Neuron* 102:128-142.
- Kawakami M, Terasawa E, Kawachi J (1964) Studies on the oxytocin sensitive component in the reticular activating system. *Jpn J Physiol* 14:102-121.
- Kecskés M, Henn-Mike N, Agócs-Laboda Á, Szócs S, Petykó Z, Varga C (2020) Somatostatin expressing GABAergic interneurons in the medial entorhinal cortex preferentially inhibit layer III-V pyramidal cells. *Communications Biology* 3:754.
- Ohara S, Rannap M, Tsutsui KI, Draguhn A, Egorov AV, Witter MP (2023) Hippocampal-medial entorhinal circuit is differently organized along the dorsoventral axis in rodents. *Cell Reports* 42:112001.
- Ohara S, Onodera M, Simonsen ØW, Yoshino R, Hioki H, Iijima T, Tsutsui K, Witter MP (2018) Intrinsic projections of layer Vb neurons to layers Va, III, and II in the lateral and medial entorhinal cortex of the rat. *Cell Reports* 24:107-116.
- Qin H, Fu L, Hu B, Liao X, Lu J, He W, Liang S, Zhang K, Li R, Yao J, Yan J, Chen H, Jia H, Zott B, Konnerth A, Chen X (2018) A visual-cue-dependent memory circuit for place navigation. *Neuron* 99:47-55.
- Sargolini F, Fyhn M, Hafting T, McNaughton BL, Witter MP, Moser MB, Moser EI (2006) Conjunctive representation of position, direction, and velocity in entorhinal cortex. *Science* 312:758-762.
- Sato MT, Tokunaga A, Kawai Y, Shimomura Y, Tano Y, Senba E (2000) The effects of binocular suture and dark rearing on the induction of c-fos protein in the rat visual cortex during and after the critical period. *Neuroscience Research* 36:227-233.

- Shang C, Chen Z, Liu A, Li Y, Zhang J, Qu B, Yan F, Zhang Y, Liu W, Liu Z, Guo X, Li D, Wang Y, Cao P (2018) Divergent midbrain circuits orchestrate escape and freezing responses to looming stimuli in mice. *Nature Communications* 9:1232.
- Solomon EA, Lega BC, Sperling MR, Kahana MJ (2019) Hippocampal theta codes for distances in semantic and temporal spaces. *Proceedings of the National Academy of Sciences of the United States of America* 116:24343-24352.
- Stensola H, Stensola T, Solstad T, Froland K, Moser MB, Moser EI (2012) The entorhinal grid map is discretized. *Nature* 492:72-78.
- Strange BA, Witter MP, Lein ES, Moser EI (2014) Functional organization of the hippocampal longitudinal axis. *Nature Reviews Neuroscience* 15:655-669.
- Sun N, Cassell MD, Perlman S (1996) Anterograde, transneuronal transport of herpes simplex virus type 1 strain H129 in the murine visual system. *Journal of Virology* 70:5405-5413.
- Sürmeli G, Marcu DC, McClure C, Garden D, Pastoll H, Nolan MF (2015) Molecularly Defined Circuitry Reveals Input-Output Segregation in Deep Layers of the Medial Entorhinal Cortex. *Neuron* 88:1040-1053.
- Tsoi SY, Öncül M, Svahn E, Robertson M, Bogdanowicz Z, McClure C, Sürmeli G (2022) Telencephalic outputs from the medial entorhinal cortex are copied directly to the hippocampus. *Elife* 11.
- Tu L, Talbot A, Gallagher NM, Carlson DE (2022) Supervising the Decoder of Variational Autoencoders to Improve Scientific Utility. *Ieee Transactions On Signal Processing* 70:5954-5966.
- van Groen T (2001) Entorhinal cortex of the mouse: cytoarchitectonical organization. *Hippocampus* 11:397-407.
- Wang Q, Burkhalter A (2007) Area map of mouse visual cortex. *Journal of Comparative Neurology* 502:339-357.
- Zemanick MC, Strick PL, Dix RD (1991) Direction of transneuronal transport of herpes simplex virus 1 in the primate motor system is strain-dependent. *Proceedings of the National Academy of Sciences of the United States of America* 88:8048-8051.

REVIEWER COMMENTS

Reviewer #1 (Remarks to the Author):

My comments were largely addressed however I still have some outstanding thoughts that the authors may find useful.

1- I strongly agree with reviewer 2 about the importance of monitoring V2->ECL5a pathway activity during the behavioural experiments. In response, the authors recorded from L5a neurons that receive V2 inputs but this is not the same as looking at the activity of the pathway. To address the comment, they should have recorded from the axons of V2 neurons in L5a. This experiment would also a lot more strongly demonstrate that visual information reaches the EC from V2 since their cFOS experiments only showed positive results under specific and strong light stimulation condition and not in a behaviourally relevant context.

3- Re: fig8m-p The increase in the activity of L5a neurons over learning might be because of enhanced synchronicity between hippocampus and deep EC that has been shown to increase with learning (DOI: 10.1016/j.cub.2023.09.011)Though the result is interesting, it should not be interpreted as a feature of V2->EC projections. To be able to make that conclusion the authors would have to record activity from axons of V2 neurons in the EC. I would suggest that the authors consider alternative explanations here.

4-Please include injection site images for the eNpHR3.0 virus in V2L or V2M. In figure 8f it would be good to show a larger part of the slice so the damage to the tissue and off target inhibition could be assessed from the distribution of axons in nearby targets.

5-I still don't know when the probe test was given to the mice in the MWM test in figure 8b. Please clarify this in the text.

Reviewer #2 (Remarks to the Author):

The authors have largely resolved my concerns, but one more suggestion remains: in figure 8m and o, could they also indicate the timepoint when the mouse reaches the platform in the calcium traces?

Response to Reviewers

We are grateful to the reviewers for their further suggestions on this study. We have revised our manuscript and added new data to improve the presentation and strengthen the conclusions of our work. Fig. 7-8 and the associated supplementary figures were reorganized to reflect the changes. A point-by-point response to reviewers' concerns is provided below. The reviewers' comments are pasted verbatim below in italic font. Our responses are in blue font. Changes to the manuscript are highlighted in red font, which are also pasted below for the convenience of the reviewers. Additionally, line numbers and figure numbers cited below are those in the newly revised manuscript unless otherwise indicated.

Reviewer #1:

My comments were largely addressed however I still have some outstanding thoughts that the authors may find useful.

1- I strongly agree with reviewer 2 about the importance of monitoring V2->ECL5a pathway activity during the behavioural experiments. In response, the authors recorded from L5a neurons that receive V2 inputs but this is not the same as looking at the activity of the pathway. To address the comment, they should have recorded from the axons of V2 neurons in L5a. This experiment would also a lot more strongly demonstrate that visual information reaches the EC from V2 since their cFOS experiments only showed positive results under specific and strong light stimulation condition and not in a behaviourally relevant context.

We agree with the reviewer's perspective. To address the reviewer's concern, we have conducted additional experiments to monitor the activity of the axons of V2 neurons in the MEC during navigation tasks. The results demonstrate that visual information reaches the MEC from V2 in spatial navigation (**Fig. 8h-8p** and **Supplementary Fig. 12d-12f**).

Lines 404-415:

Next, to directly monitor visual inputs to the MEC during navigation, we utilized fiber photometry to record the activity of V2 axons within the MEC. After injecting AAV2/9-CaMKII α -GCaMP6m into either V2M or V2L and implanting optical fiber above the vCE (**Fig. 8h-8i** and **Supplementary Fig. 12d**), we first ensured that robust Ca²⁺ responses could be recorded from the V2 axons by applying a brief light stimulus to freely moving mice (**Fig. 8j-8k**). We then recorded Ca²⁺ signals from V2 axons in mice during the BM task (**Fig. 8l**). As the training progressed for the first 4 days, the Ca²⁺ signals from both V2M and V2L axons increased in tandem with the marked improvement in locating the escape hole swiftly (**Fig. 8m-8p** and **Supplementary Fig. 12e-12f**). Furthermore, in darkness on day 5, when the activity of V2M and V2L axons decreased, the animals' ability to efficiently locate the escape hole also declined (**Fig. 8m-8p** and **Supplementary Fig. 12e-12f**). These results indicate that sustained visual inputs delivered via the V2→MEC L5a pathway are indispensable for navigation tasks. To summarize, the behavioral data so far together reveal a critical role of the V2→MEC L5a pathway in memory-guided spatial navigation.

Fig. 8h-8p:

Lines 1134-1147:

h, Schematic illustrating viral injection strategy and fiber photometry recording. Ca^{2+} transients were recorded from V2M or V2L projections in vCE of freely moving mice in a dark box. The light stimuli were delivered by a remote-controlled lamp.

i, Representative image showing optical fiber placement and GCaMP-expressing fibers of V2M or V2L in vCE.

j-k, Left: heatmaps showing the Ca^{2+} signals of V2M fibers (**j**) or V2L fibers (**k**) evoked by light stimuli (5 trials). Middle: averaged Ca^{2+} transients of mice in different groups; thick lines indicate mean and shaded areas indicate SEM. Right: summary plot exhibiting the peak of Ca^{2+} signals of V2M fibers (**j**, $n = 8$ mice) or V2L fibers (**k**, $n = 9$ mice).

l, Schematic illustrating fiber photometry recording during BM tests. Cyan circle denotes the escape hole.

m and **o**, Mean Ca^{2+} signal of V2M fibers (**m**, $n = 8$ mice) or V2L fibers (**o**, $n = 9$ mice) on training days 1-4 and in the dark condition on day 5. Shaded areas indicate SEM. Blue arrowheads indicate the starting timepoint of the escape activity of mice.

n and **p**, Plot of the escape latencies (black) and their corresponding amplitudes of Ca^{2+} signals (green) at different training days (for V2M, $n = 8$ mice; for V2L, $n = 9$ mice).

Supplementary Fig. 12d-12f:

Lines 1374-1377:

d, Confocal images showing GCaMP viral injection site within V2M (left) and V2L (right). Scale bars, 200 μm .

e-f, Examples of the movement trajectory (top) and the corresponding Ca^{2+} signals (bottom) of V2M fibers (**e**) or V2L fibers (**f**) on training days 1-4 and in the dark condition on day 5. The Ca^{2+} signal traces and their corresponding escape paths are marked in red.

3- Re: *fig8m-p* The increase in the activity of L5a neurons over learning might be because of enhanced synchronicity between hippocampus and deep EC that has been shown to increase with learning (DOI: 10.1016/j.cub.2023.09.011) Though the result is interesting, it should not be interpreted as a feature of V2- \rightarrow EC projections. To be able to make that conclusion the authors would have to record activity from axons of V2 neurons in the EC. I would suggest that the authors consider alternative explanations here.

We thank the reviewer for providing this valuable information. The study from Santos-Pata et al. did show that dMEC-hippocampal coordination increased with learning, but it also reported that the coordination happened only during “rest” periods rather than “run” periods. Therefore, we think that the learning-related increase in the activity of L5a neurons during navigation tasks cannot be simply attributed to dMEC-hippocampal coordination, if at all. Nonetheless, we agree with the reviewer’s opinion that alternative mechanism may have contributions. Thus, we have cited the paper of Santos-Pata et al. and revised the text to call attention of the audience. We also agree with the reviewer’s opinion that activity from axons of V2 neurons should be recorded to corroborate the feature of V2 \rightarrow EC projections and have added these data (see our response to Point 1).

Lines 389-393:

The foregoing results corroborate the critical role of MEC L5a neurons receiving V2 projections during navigation. However, since MEC L5a also receives inputs from other brain areas, the above observations may be attributed to alternative mechanisms, such as learning-related increase in

coordination between the deep MEC and the hippocampus⁴⁵. Thus, we would like to investigate features of V2→MEC L5a projections during navigation tasks directly. We first used an optogenetic approach to inquire into ...

Lines 834-835:

45. Santos-Pata, D., Barry, C. & Ólafsdóttir, H.F. Theta-band phase locking during encoding leads to coordinated entorhinal-hippocampal replay. *Current Biology* **33**, 4570-4581.e4575 (2023).

4-Please include injection site images for the eNpHR3.0 virus in V2L or V2M. In figure 8f it would be good to show a larger part of the slice so the damage to the tissue and off target inhibition could be assessed from the distribution of axons in nearby targets.

Thanks for the suggestions. The injection site of the eNpHR3.0 virus in V2L or V2M are now displayed in **Supplementary Fig. 12a**. Additionally, new **Fig. 8b** (previous **Fig. 8f**) provides a larger view of the brain slices.

Supplementary Fig. 12a:

Line 1369:

a, Confocal images showing eNpHR3.0 viral injection site within V2M (left) and V2L (right). Scale bars, 200 μ m.

Fig. 8b:

5-I still don't know when the probe test was given to the mice in the MWM test in figure 8b. Please clarify this in the text.

We are sorry that the original manuscript did not express it clearly. After 6 days of training, the probe test was given on day 7. We have expressed the time explicitly in the newly revised manuscript.

Lines 356-358:

During training sessions (days 1-6) in MWM, there was no difference in learning ability between the two groups of mice (**Supplementary Fig. 10a**). During the probe test (day 7) in MWM, ...

Reviewer #2:

The authors have largely resolved my concerns, but one more suggestion remains: in figure 8m and o, could they also indicate the timepoint when the mouse reaches the platform in the calcium traces?

Thanks for the suggestion. We have added arrowheads to indicate the timepoint when the mouse reached the platform in those figures (**Fig. 7m and 7o** in the revised manuscript).

Fig. 7m and 7o:

Lines 1104-1106:

m and **o**, Examples of Ca^{2+} signals of vCE L5a neurons receiving V2M projections (**m**) or V2L projections (**o**) on training days 1- 4 and in the 1-s lighting or the dark condition on day 5. **Arrowheads indicate the timepoint when the mouse reached the platform.**

REVIEWER COMMENTS

Reviewer #1 (Remarks to the Author):

First of all I congratulate the authors for swiftly providing extra data on difficult experiments. I value their commitment for improving the manuscript.

Thank you for providing extra experiments on recordings from the fiber tracks during BM. Can you clarify when the bright light is put on in relation to the traces displayed in 8m and o?. I have looked at the methods and I understand that a transparent start box was used. I have no issues with the experiment if the aversive light stimulus was switched on for the whole duration of the recording. If it was switched on just when the animal was released from the start box, then the signal you pick could merely be the result of the bright light stimulus and has nothing to do with the task itself. In other words you would be repeating the same experiment you presented in figure 8j and k.

Also please specify the duration of light stimulation in Figure 8j and k.

Reviewer #2 (Remarks to the Author):

The arrowheads added in figure 5 are too small to see. The strange thing is that one or two arrowheads indicate the ending timepoints for all the traces?

Response to Reviewers

We sincerely appreciate the reviewers for their further comments on this study. We have improved the clarity of experimental methods and figure representation. A point-by-point response to reviewers' concerns is provided below. The reviewers' comments are pasted verbatim below in italic font. Our responses are in blue font. Changes to the manuscript are highlighted in red font, which are also pasted below for the convenience of the reviewers.

Reviewer #1:

First of all I congratulate the authors for swiftly providing extra data on difficult experiments. I value their commitment for improving the manuscript.

Thank the reviewer for the positive feedback and acknowledgment of our commitment for improving the manuscript. We appreciate the reviewer's thorough review and insightful comments.

Thank you for providing extra experiments on recordings from the fiber tracks during BM. Can you clarify when the bright light is put on in relation to the traces displayed in 8m and o?. I have looked at the methods and I understand that a transparent start box was used. I have no issues with the experiment if the aversive light stimulus was switched on for the whole duration of the recording. If it was switched on just when the animal was released from the start box, then the signal you pick could merely be the result of the bright light stimulus and has nothing to do with the task itself. In other words you would be repeating the same experiment you presented in figure 8j and k.

We appreciate the reviewer's careful consideration to this important aspect. It was also our concern. Thus, we utilized a transparent cup to avoid abrupt light change. The light was switched on for the entire duration of the recording (except for trials in dark on day 5). It is also crucial to note that the Ca²⁺ signals from V2 axons increased as the training progressed (Fig. 8m-8p). These results indicate that the signals are not responses induced by brief light stimulus like those in Fig. 8j-8k.

We have revised the text to explain our purpose of using a transparent cup.

Lines 704-705:

For each trial, mice were first placed in an opaque cup (for **Fig. 8e**) or a transparent cup (for **Fig. 8l to avoid abrupt light change**) in the center of the maze for 5 s.

Also please specify the duration of light stimulation in Figure 8j and k.

We have added the information in the manuscript.

Lines 407-408:

... by applying a brief 2-second light stimulus to freely moving mice (**Fig. 8j-8k**).

Reviewer #2:

The arrowheads added in figure 5 are too small to see. The strange thing is that one or two arrowheads indicate the ending timepoints for all the traces?

We appreciate the careful examination by the reviewer. We understand that the reviewer wanted to refer to Fig. 7. The arrowheads in Fig. 7 do not indicate the ending timepoints for all the traces, but specifically for the traces of the same color with the arrowheads. In the examples shown in Fig. 7m and 7o, ending timepoints were only indicated for the marked traces, as mice did not reach the platform within 15 seconds in the remaining traces. We are sorry that the arrowheads were not clear enough. We have increased the size and changed them to arrows.

Fig. 7m and 7o:

Lines 1104-1106:

m and **o**, Examples of Ca^{2+} signals of vCE L5a neurons receiving V2M projections (**m**) or V2L projections (**o**) on training days 1- 4 and in the 1-s lighting or the dark condition on day 5. **Colored arrows** indicate the timepoint when the mouse reached the platform **on the trace of the same color**.

REVIEWERS' COMMENTS

Reviewer #1 (Remarks to the Author):

Thank you for the clarification. I have no further comments.

Reviewer #2 (Remarks to the Author):

I have no further comments and suggestions.

Response to Reviewers

Reviewer #1:

Thank you for the clarification. I have no further comments.

We are delighted to know the reviewer was satisfied with our revision. We truly appreciate the time and effort the reviewer dedicated throughout the review process.

Reviewer #2:

I have no further comments and suggestions.

We are delighted to know the reviewer was satisfied with our revision. We truly appreciate the time and effort the reviewer dedicated throughout the review process.